# Interferon Regulatory Factor 3-Mediated Signaling Limits Middle-East Respiratory Syndrome (MERS) Coronavirus Propagation in Cells from an Insectivorous Bat

**DOI:** 10.3390/v11020152

**Published:** 2019-02-13

**Authors:** Arinjay Banerjee, Darryl Falzarano, Noreen Rapin, Jocelyne Lew, Vikram Misra

**Affiliations:** 1Department of Veterinary Microbiology, Western College of Veterinary Medicine, University of Saskatchewan, Saskatoon, SK S7N 5B4, Canada; banera9@mcmaster.ca (A.B.); darryl.falzarano@usask.ca (D.F.); noreen.rapin@usask.ca (N.R.); 2Vaccine and Infectious Disease Organization-International Vaccine Centre (VIDO-Intervac), University of Saskatchewan, Saskatoon, SK S7N 5E3, Canada; jocelyne.lew@usask.ca

**Keywords:** bat, IRF3, MERS-CoV, interferon

## Abstract

Insectivorous bats are speculated to be ancestral hosts of Middle-East respiratory syndrome (MERS) coronavirus (CoV). MERS-CoV causes disease in humans with thirty-five percent fatality, and has evolved proteins that counteract human antiviral responses. Since bats experimentally infected with MERS-CoV do not develop signs of disease, we tested the hypothesis that MERS-CoV would replicate less efficiently in bat cells than in human cells because of its inability to subvert antiviral responses in bat cells. We infected human and bat (*Eptesicus fuscus*) cells with MERS-CoV and observed that the virus grew to higher titers in human cells. MERS-CoV also effectively suppressed the antiviral interferon beta (IFNβ) response in human cells, unlike in bat cells. To determine if IRF3, a critical mediator of the interferon response, also regulated the response in bats, we examined the response of IRF3 to poly(I:C), a synthetic analogue of viral double-stranded RNA. We observed that bat IRF3 responded to poly(I:C) by nuclear translocation and post-translational modifications, hallmarks of IRF3 activation. Suppression of IRF3 by small-interfering RNA (siRNA) demonstrated that IRF3 was critical for poly(I:C) and MERS-CoV induced induction of IFNβ in bat cells. Our study demonstrates that innate antiviral signaling in *E. fuscus* bat cells is resistant to MERS-CoV-mediated subversion.

## 1. Introduction

Bats are ecologically important mammals that are speculated to be reservoirs of several emerging viruses, including coronaviruses [1,2]. Bats have been called global reservoirs of deadly coronaviruses (CoVs) [3] and over 200 different viruses have been isolated or detected in bats [1], with new viruses being detected on a regular basis. Over nine hundred coronavirus sequences from bats have been reported [4] and recently, coronaviruses thought to have spilled-over from bats have caused serious disease in humans and agricultural animals. These include severe acute respiratory syndrome (SARS)-CoV, Middle-East respiratory syndrome (MERS)-CoV, porcine epidemic diarrhea virus (PEDV), and swine acute diarrhea syndrome (SADS) coronavirus [5,6,7,8,9,10,11,12]. The spill-over of these viruses to susceptible hosts, including humans and agricultural animals, often results in severe disease.

MERS-CoV causes severe respiratory illness in humans, including a cough, fever, and shortness of breath. Since September 2012, the World Health Organization has been notified of 2249 laboratory-confirmed cases of MERS-CoV infection, with at least 798 deaths. Twenty-seven countries have reported cases of MERS-CoV [13]. SARS-CoV caused a pandemic in 2003–2004 that killed ten percent of the over 8000 infected individuals [7] and PEDV was responsible for over $300 million in losses to the US swine industry in 2013–2014 [14]. In many cases, infections with SARS- and MERS-CoVs lead to severe illness, including death, as only supportive care is available.

Insectivorous vespertilionid bats from the suborder Yangochiroptera have been proposed as an ancestral source of MERS-CoV as several closely related coronaviruses have been described in these bats [8,15,16,17,18]. Although virus-host interactions for MERS-CoV are being studied in human cells [19,20,21,22,23], little information exists on how MERS-CoV interacts with cells from an ancestral and/or potential reservoir host, such as bats. Moreover, we do not understand the type of innate immune response that is mounted by bat cells against MERS-CoV.

The innate immune response is the first line of defense against invading pathogens, including viruses. When the pathogen-associated molecular patterns (PAMPs) of a virus are sensed by a cell, the cell activates various signaling pathways to counteract the ensuing infection. During coronavirus infection and subsequent virus replication, the cell recognizes viral nucleic acid through cellular receptors or pattern recognition receptors (PRRs) and mounts an antiviral response. This response is mediated through interferon regulatory factor 3 (IRF3), a key transcription factor involved in antiviral interferon signaling. IRF3 exists as a monomer in the cytoplasm of human cells. When PRRs are activated in a cell in response to viral nucleic acids, cellular kinases such as TANK-binding kinase 1 (TBK1) phosphorylate IRF3, which then dimerizes and translocates to the nucleus of the cell. Once in the nucleus, phosphorylated IRF3 binds to its response elements in the promoters of antiviral interferon genes and enhances the expression of interferons, such as interferon beta (IFNβ) (reviewed here [24]).

Interferons are key molecules that activate interferon stimulated genes (ISGs) such as 2′-5′-oligoadenylate synthetase 1(OAS1) and interferon-induced GTP-binding protein Mx1 in infected and neighboring cells (reviewed here [25]). ISGs inhibit virus replication in these cells through various mechanisms [26]. For example, OAS1 inhibits virus replication by selectively degrading viral RNA in combination with RNase L [27]. 

Viruses have evolved different mechanisms to counteract the antiviral interferon responses (reviewed here [28,29]). Like other viruses, coronaviruses such as SARS-CoV, MERS-CoV, and PEDV have evolved different strategies to inhibit interferon signaling in host cells [23,30,31,32]. MERS-CoV structural and accessory proteins inhibit interferon production in human cells, predominantly through the inhibition of IRF3-mediated signaling [23]. However, we do not know if MERS-CoV can subvert interferon signaling in bat cells in a similar manner.

Most studies on MERS-CoV-host interactions have been performed in human cells by ectopically expressing MERS-CoV proteins. Little research has been done to study the virus in human cells, largely due to containment requirements. To our knowledge, there are no studies that have looked at the effect of MERS-CoV infection on IRF3-mediated innate immune responses in bat cells. Since vespertilionid bats have been speculated as evolutionary hosts of MERS-CoV [8,15,33,34], we tested the hypothesis that MERS-CoV cannot effectively shut-down interferon responses in cells from these bats. To test our hypothesis, we infected big brown bat (vespertilionid bat; *Eptesicus fuscus*) kidney [35] and human lung cells with MERS-CoV. We observed that MERS-CoV propagated to higher levels in human cells than bat cells. Consistent with other studies, human cells infected with MERS-CoV did not express transcripts for IFNβ. In contrast, when Efk3 cells from big brown bats were infected with MERS-CoV, they expressed robust amounts of IFNB transcripts. Moreover, small interfering RNA (siRNA) mediated knock-down of and CRISPR/Cas9-mediated deletion of IRF3 demonstrated that IRF3 is critical for poly(I:C) and MERS-CoV-induced antiviral IFNβ response in bat cells.

Our data suggest that the IRF3 signaling pathway in big brown bat cells is resistant to virus-mediated subversion and is critical to limiting MERS-CoV replication. This and other antiviral signaling pathways warrant further investigation to identify adaptations that allow bats to mount an antiviral response to a viral infection. Understanding cellular and molecular virus-host interactions in reservoir hosts may help in predicting factors that contribute to the emergence of these viruses from their natural host.

## 2. Materials and Methods

### 2.1. Cell Culture and Virus Infection

*Eptesicus fuscus* kidney cells [35] (Efk3 or bat cells) were grown in Dulbecco’s Minimal Essential Medium with GlutaGro (DMEM; Corning, New York, NY, USA) containing 10% fetal bovine serum (FBS; Sigma, Milwaukee, WI, USA), penicillin/streptomycin (Gibco, Gaithersburg, MD, USA), and 1% GlutaMax (Gibco). Human lung (MRC5) cells (ATCC CCL-171) were cultured in Minimum Essential Medium Eagle (MEM; Corning) supplemented with 10% FBS, 1/100 non-essential amino acids (NEAA; Gibco), 1/100 4-(2-hydroxyethyl)-1-piperazineethanesulfonic acid (HEPES; Gibco), and 1/1000 gentamycin (Gibco). Vero (green monkey kidney) cells were grown in DMEM supplemented with 10% FBS and penicillin/streptomycin. A549 cells (ATCC CCL-185) were grown in F12-K medium (Gibco) with 10% FBS and penicillin/streptomycin. Huh7 cells (gift from Dr. Ralf Bartenschlager, Heidelberg University) were grown in DMEM supplemented with 10% FBS and penicillin/streptomycin. CaLu3 cells (ATCC HTB-55) were grown in MEM supplemented with 10% FBS and penicillin/streptomycin. Tb1-Lu cells (ATCC CCL-88; gift from Drs. Heidi Hood and Amrit Boese) were grown in DMEM supplemented with 10% FBS, GlutaMax and penicillin/streptomycin. Cells were incubated in a humidified incubator at 37 °C with 5% CO_2_.

For virus infection studies, cells were seeded at a concentration of 3 × 10^5^ cells/well in a six-well plate. Based on the experiment (refer to results), the cells were infected with varying multiplicity of infection (MOI) of MERS-CoV (strain EMC/2012) in a containment level 3 laboratory. After 1 h, the inoculum was removed, cells were rinsed three times with media to remove residual inoculum, and fresh complete medium was added on the cells.

### 2.2. Virus Titration

MERS-CoV virus infections and titrations were done in a containment level 3 laboratory. For titrating the amount of virus in supernatants from infected cells, Vero cells were seeded in 96-well plates at a concentration of 10^5^ cells/well in 100 μL of complete media. The plates were incubated at 37 °C overnight. The next day, media was taken off the cells and 50 μL of 1:10 serially diluted virus containing supernatant was added to the plates. The plates were incubated at 37 °C for 1 h. After incubation, the virus containing supernatant was discarded and 100 μL of complete media was added to the plates. The plates were incubated at 37 °C for three and five days, respectively. A cytopathic effect was observed under a microscope. A tissue culture infectious dose of 50/mL (TCID_50_/mL) was calculated using the Spearman and Karber algorithm [36,37].

### 2.3. TLR3 Stimulation

MRC5 and Efk3 cells were seeded at a concentration of 3 × 10^5^ cells/well in six-well plates and transfected with 750 ng/mL poly(I:C) (InvivoGen, San Diego, CA, USA) using Lipofectamine 2000 (Invitrogen, Camarillo, CA, USA) as previously described [38]. Briefly, 750 ng/mL poly(I:C) was mixed in a total volume of 250 μL of TransfectaGro (Corning) and 12 μL of lipofectamine 2000. This mixture was incubated at room temperature for 15 min and added to cells in complete medium. Cells were harvested 16 h post-transfection and RNA was extracted.

### 2.4. Nucleic Acid Extraction, qRT-PCR, and Conventional PCR

All RNA extractions were performed using the RNeasy Plus Mini kit (QIAGEN, Hilden, Germany) as per the manufacturer’s instructions. cDNA was prepared using the iScript gDNA clear kit (Bio-Rad, Hercules, CA, USA) as per the manufacturer’s instructions. A total of 500 ng of RNA was used for cDNA preparation. cDNA was used as a template for the quantification of target genes. Genomic DNA was extracted using the DNeasy blood and tissue kit (QIAGEN) as per the manufacturer’s instructions.

qRT-PCR assays targeting respective cellular genes and the normalizer (Glyceraldehyde-3-phosphate; GAPDH) were performed for both MRC5 and Efk3 cells. Primer sequences for human and bat genes have been published before [38]. Primer sequences for dipeptidyl-peptidase 4 (DPP4) were obtained from a preprint on Bioarchive [39]. Bio-Rad’s CFX96 Touch PCR thermocycler was used in conjunction with Bio-Rad’s Ssofast Evagreen supermix (Bio-Rad) and samples were prepared as previously mentioned [40]. For qRT-PCR, after the initial denaturation step of 95 °C for 5 min, two-step cycling for 40 cycles was performed at 95 °C/10 s and 56 °C/30 s. Absorbance readings were acquired after each cycle. The final three steps were carried out at 95 °C/1 min, 55 °C/30 s, and 95 °C/30 s to generate the dissociation curve. Absorbance readings for the dissociation curve were acquired at every degree from 55–95 °C. Relative fold change in gene expression between the two groups of cells (treated/infected and mock treated/infected) was calculated after normalizing the Ct values using GAPDH. A difference of one Ct indicates a two-fold difference in gene expression. Primer sequences can be found in Table 1 and here [38]. 

Conventional PCR was carried out to amplify the first exon of the big brown bat IRF3 gene to detect deletion by CRISPR/Cas9. Primer sequences are listed below. PCR was performed using the following thermal cycle profile: initial denaturation for 3 min at 94 °C, 35 PCR cycles at 94 °C/30 s, 56 °C/30 s, and 72 °C/1 min. The final extension was at 72 °C for 10 min.

### 2.5. Agarose Gel Electrophoresis

One percent agarose (Invitrogen, USA) gels were prepared using 0.5× TBE (Tris—1M (VWR), Ethylenediaminetetraacetic acid disodium salt (EDTA) solution—0.02 M (Gibco), and Boric acid—1M; pH 8.4). A 1 μL SYBR Safe DNA gel stain (Invitrogen, Camarillo, CA, USA) was added for every 1 mL of gel. In total, 10 μL of PCR or qRT-PCR products were electrophoresed on the gel for 1 h at 105 volts and visualized under an ultraviolet gel imaging system (AlphaImager HP).

### 2.6. Knock-Down of IRF3 Transcripts in Efk3 and MRC5 Cells

Dicer-ready siRNA (DsiRNA) specific to big brown bat and human IRF3 were designed and obtained through Integrated DNA Technologies (IDT). A 100 nM final concentration of a 1:1 mixture of two DsiRNAs per cell line (Table 1) targeting separate regions on the big brown bat and human IRF3 transcript were transfected into Efk3 and MRC5 cells using Lipofectamine 2000. Scrambled non-specific DsiRNA (NC DsiRNA; IDT) was used as a negative control.

### 2.7. Generating IRF3 Knockout Bat Cells

Efk3 cells were seeded in a 24-well plate at a concentration of 9 × 10^4^ cells/well and the manufacturer’s (Invitrogen) recommended protocol was followed to generate IRF3 knockout cells. Briefly, the cells were transfected with 7.5 pmol of guide RNA (gRNA; crRNA:tracrRNA duplex; Invitrogen) using Lipofectamine Cas9 Plus Reagent (Invitrogen) in combination with 1290 ng of the Cas9 protein (Invitrogen). The cells were transfected twice on different occasions to increase the transfection efficiency. Two separate crRNAs were used to generate two types of gRNAs: crRNA-2- AUG UCG GGC CUG CUA ACA AU (Direction “−”; IDT) and crRNA-3- CAU UGU UAG CAG GCC CGA CA (Direction “+”; IDT). After allowing the cells to grow till confluency, a 607 bp region in the exon that was targeted by the crRNA for deletion was amplified and sequenced (Macrogen, Seoul, Korea). Primers IRF3-CRISPR-gDNA-F3: AGGCTTTCTGTGGGGGATTG and IRF3-CRISPR-gDNA-R3: AGATGCCAAAGTCCTCCTG were used to amplify the 607 bp locus in the genomic DNA by conventional PCR. Deletions were confirmed by sequencing (Macrogen) and the sequencing results were analyzed using an online tool (Tracking of Indels by Decomposition or TIDE) [41] to determine the indel spectrum (frequency of targeted mutations generated in a pool of cells) in the target sequence. The cell population with the highest CRISPR knockout efficiency was cloned by end-point dilution to obtain single cell colonies. Thirty-five single cell colonies were scaled up and knockout cell lines were subsequently selected by performing immune blots to detect IRF3 protein expression. Three knockout cell lines were obtained. cr2-9 and cr2-12 were generated by using crRNA-2 and cr3-8 was generated using crRNA-3. Cr3-8, also called Efk3-cr3-8, was used for subsequent experiments.

### 2.8. Immunofluorescence

Efk3 and MRC5 cells were seeded at a concentration of 3 × 10^5^ cells/well in six-well plates (Thermo Scientific) with glass cover-slips. Cells were treated with 750 μg/mL poly(I:C) after 24 h and incubated for another 16 h. Media was discarded and cells were rinsed with 2 mL PBS. Cover-slips were transferred to wells containing ice-cold methanol in six-well plates and incubated for 20 min in a freezer. Methanol was discarded and cells were washed with PBS. Cells were blocked using a blocking solution (PBS, 10% donor calf serum (Sigma) and 0.1% Tween 20 (USB)). Primary staining for IRF3 and GAPDH was performed using 1:100 dilution of rabbit anti-IRF3 (Abcam; Catalogue number: ab68481; RRID: AB_11155653) and mouse anti-GAPDH (EMD Milipore; Catalogue number: AB2302; RRID: AB_10615768). Secondary staining was performed using 4 μg/mL goat anti-mouse Alexa 488 (Molecular Probes; Catalogue number: A-11001; RRID: AB_2534069), 0.1 μg/mL goat anti-rabbit Cy5 (GE Healthcare; Catalogue number: PA45012; RRID: AB_772204), and 0.2 μg/mL Hoechst 33342 (Molecular Probes; Catalogue number: H3570) in blocking solution. Cells were observed under a TCS SP5 confocal microscope (Leica, Allendale, NJ, USA). Mean fluorescence was measured using Image J (Version 1.49) and calculated using a formula previously described [42].

### 2.9. Cell Fractionation

MRC5 and Efk3 cells were seeded at a concentration of 3 × 10^5^ cells/100 mm plate in 10 mL of media. Twenty-four hours after the cells were seeded, cells were mock transfected or transfected with 750 ng/mL poly(I:C) using Lipofectamine 2000. Twelve hours after transfection, cells from two 100 mm plate/per treatment type were trypsinized and pooled. Cell fractionation was carried out as per manufacturer’s recommendation using the NE-PER nuclear and cytoplasmic extraction kit (ThermoScientific, San Diego, CA, USA).

### 2.10. Immune Blots

Efk3 and MRC5 cells were seeded at a concentration of 3 × 10^5^ cells/well in six-well plates and simultaneously transfected with 100 nM of 1:1 cocktail of two different siRNA specific to IRF3 (Table 1) and NC siRNA. Cells were transfected with 750 ng/mL poly(I:C) using lipofectamine 2000 (Invitrogen) or mock transfected with lipofectamine 2000. Cells were harvested in sample buffer for immune blots 48 h post transfections. Immune blots were carried out as previously mentioned [43]. Briefly, samples were denatured in a reducing sample buffer and run on a reducing gel. Proteins were blotted from the gel onto polyvinylidene difluoride (PVDF) membranes and detected using primary and secondary antibodies. Primary antibodies used were: 1:1000 mouse anti-GAPDH (EMD Milipore; Catalogue number: AB2302; RRID: AB_10615768), 1:1000 rabbit anti-IRF3 (Abcam; Catalogue number: ab68481; RRID: AB_11155653), 1:1000 rabbit anti-calnexin (Santa Cruz Biotechnology; Catalogue number: sc-11397; RRID: AB_2243890), and 1:1000 rabbit anti-Lamin B1 (Abcam; Catalogue number: ab16048; RRID: AB_10107828). Secondary antibodies used were: 1:10,000 goat anti-mouse Alexa 488 (Molecular Probes; Catalogue number: A-11001; RRID: AB_2534069) and 1:10,000 goat anti-rabbit Cy5 (GE Healthcare; Catalogue number: PA45012; RRID: AB_772204). Blots were observed and imaged using a Typhoon Scanner (Amersham Biosciences, Waukesha, WI, USA).

### 2.11. Phylogenetic Analysis

Mammalian IRF3 nucleotide sequences were obtained from the National Centre for Biotechnology Information’s (NCBI) database (Table 2). The evolutionary history was inferred by using the Maximum Likelihood method (1000 Bootstrap) based on the Tamura-Nei model [44]. The tree with the highest log likelihood is shown. The percentage of trees in which the associated taxa cluster together is shown next to the branches. Initial tree(s) for the heuristic search were obtained automatically by applying Neighbor-Join and BioNJ algorithms to a matrix of pairwise distances estimated using the Maximum Composite Likelihood (MCL) approach, and then selecting the topology with a superior log likelihood value. The tree is drawn to scale with branch lengths measured in the number of substitutions per site. The analysis involved 30 nucleotide sequences. All positions containing gaps and missing data were eliminated. There were a total of 1213 positions in the final dataset. Evolutionary analyses were conducted in MEGA7 [45].

### 2.12. Statistics

Significance of the data was determined by a two-tailed Mann Whitney *U* test for non-parametric independent samples using IBM SPSS (Version 21).

## 3. Results

### 3.1. MERS-CoV Propagates to Lower Levels in Big Brown Bat Cells

Since there are few studies on the interactions of MERS-CoV with the innate immune responses in insectivorous bats, we compared the dynamics and consequence of MERS-CoV infection in *E. fuscus* (big brown bat) kidney (Efk3) and human lung (MRC5) cells. We observed that MERS-CoV propagated to significantly higher titers in human cells than big brown bat cells that were infected with either a low (Figure 1A) or high (Figure 1B) multiplicity of infection (MOI). To rule out the possibility that MRC5 cells were unique among human cell lines in supporting high levels of MERS-CoV, we infected additional human liver (Huh7) and human lung (A549 and CaLu3) cells (Figure 1C and Figure 2). We also infected insectivorous bat lung cells (Tb1-Lu; from the bat *Tadarida brasiliensis*), the only other commercially available vespertilionid bat cell line (Figure 1D and Figure 2). The virus replicated efficiently in MRC5, CaLu3, and Huh7 cells, but not in A549 cells (Figure 1C). While virus yields from the three permissive human cell lines varied, they were still higher than the yield from bat Efk3 cells. MERS-CoV did not replicate in bat lung (Tb1-Lu) cells (Figure 1D). To determine if the levels of DPP4, the putative cellular receptor for MERS-CoV, influences virus propagation in MRC5 and Efk3 cells, we quantified the amount of DPP4 transcripts in MRC5 and Efk3 cells by quantitative real time PCR (qRT-PCR). Both MRC5 and Efk3 cells had comparable levels of DPP4 transcripts (Figure 1E). The cytopathic effect (CPE) observed as a result of viral infection with an MOI of 10 infectious units/cell was also more pronounced in MRC5 cells 24 h post-infection (hpi) (Figure 1H). A similar effect was also observed at a lower MOI (0.01 infectious unit/cell), where bat cells demonstrated reduced CPE at 72 hpi compared to human cells (Figure 2). 

### 3.2. In Contrast to Human Cells, MERS-CoV Induces IFNβ Transcripts in Bat Cells

MERS-CoV, like other viruses, has evolved strategies to counteract cellular defensive responses [23,29]. To determine if MERS-CoV could efficiently shut-down antiviral interferon responses in bat cells, we infected big brown bat kidney cells (Efk3) and human lung cells (MRC5) with MERS-CoV at an MOI of 10 tissue culture infectious dose 50 (TCID_50_) units/cell. We extracted RNA from these cells at 0, 12, 24, and 48 hpi and determined the fold increase in IFNβ transcripts in infected cells relative to mock infected cells after normalization with transcripts for glyceraldehyde-3-phosphate dehydrogenase (GAPDH). We observed that MERS-CoV infection did not lead to an increase in IFNβ transcripts in MRC5 cells (Figure 1F). We were unable to detect IFNβ transcripts in human Huh7 cells infected with MERS-CoV at a high MOI. In contrast, MERS-CoV infection produced a significant increase in IFNβ transcripts at later time points of 24 and 48 hpi in Efk3 cells (Figure 1F). 

Infection with MERS-CoV is associated with a delayed but exaggerated inflammatory response [46]. We have previously shown that big brown bat cells actively suppress a strong inflammatory response when treated with poly(I:C) [38]. In this study, we also tested the ability of big brown bat cells to suppress MERS-CoV-mediated expression of TNFα, a key systemic inflammatory cytokine. We observed that MRC5 cells that were infected with MERS-CoV expressed significantly higher levels of TNFα transcripts compared to Efk3 cells (Figure 1G).

### 3.3. IRF3 Localizes in the Nucleus Of Big Brown Bat Cells in Response to Poly(I:C)

Since bat cells (Efk3) showed an increase in IFNβ transcripts after MERS-CoV infection, we hypothesized that IRF3-mediated signaling in big brown bat cells, unlike in human cells, would be resistant to MERS-CoV-mediated subversion. However, there is currently no information about the role of IRF3 in interferon signaling in bats. To test if IRF3 is critical for an IFN response in big brown bat cells, we treated Efk3 cells with poly(I:C), a synthetic double-stranded analogue of viral RNA and observed the cellular location of IRF3 by immunofluorescent microscopy. In both human (MRC5) and bat (Efk3) cells, IRF3 localized to the nucleus of the cells after poly(I:C) treatment; a hallmark of IRF3 activation (Figure 3A). We quantified the amount of nuclear IRF3 in mock and poly(I:C) treated cells. In both MRC5 and Efk3 cells, there was significantly more IRF3 in the nucleus of poly(I:C)-treated cells relative to mock treated cells (Figure 3B). We separated Efk3 and MRC5 cells treated with poly(I:C) into cytoplasmic and nuclear fractions. We performed immune blots for phosphorylated IRF3 (pIRF3), but none of the commercial antibodies that we purchased cross-reacted with the big brown bat pIRF3. However, numerous studies have shown that pIRF3, when detected with an anti-IRF3 antibody, displays several apparent higher molecular weight bands with a lower electrophoretic mobility than the unmodified IRF3 (laddering effect) [47,48,49,50]. We observed a similar laddering effect with IRF3 in the nuclear fractions of poly(I:C) treated MRC5 and Efk3 cells (Figure 3C; arrow), further supporting the translocation of IRF3 to the nucleus of poly(I:C) treated cells. In the figure, the cytoplasmic fraction is stained for calnexin, an integral protein of the endoplasmic reticulum and nuclear fractions are stained for lamin, a nuclear membrane protein. 

### 3.4. IRF3 is Critical for Antiviral Interferon beta (IFNβ) Production in Big Brown Bat Cells

To determine the role of IRF3 in interferon signaling in big brown bat cells, we knocked-down IRF3 in human (MRC5) and bat (Efk3) cells using siRNA and generated IRF3 knockout big brown bat cells (cr2-9, cr2-12, and cr3-8) using clustered regularly interspaced short palindromic repeats (CRISPR/Cas9) technology. We quantified the increase in IFNβ transcripts after poly(I:C) treatment by a quantitative real-time polymerase chain reaction (qRT-PCR). We confirmed the reduction in the expression of IRF3 in IRF3-specific siRNA-treated Efk3 and MRC5 cells and IRF3 knockout bat cells by performing immune blots (protein panel; Figure 4A,B,D). We observed that knocking-down IRF3 in both MRC5 and Efk3 cells significantly reduced the expression of IFNβ transcripts in response to poly(I:C) in these cells (1338.86 fold and 4.78 fold in MRC5 and Efk3 cells, respectively) relative to cells treated with non-specific negative control siRNA (ncsiRNA; 59,585.39 fold and 763.53 fold in MRC5 and Efk3 cells, respectively; Figure 4A,B). IRF3 knockout bat cells (Figure 4C) did not respond to poly(I:C) stimulation by upregulating IFNβ transcripts relative to wildtype bat (Efk3-WT) cells (Figure 4D). 

### 3.5. IRF3-Mediated Signaling Inhibits MERS-CoV Propagation in Big Brown Bat Cells

To determine if IRF3-mediated antiviral signaling suppressed MERS-CoV replication in big brown bat (Efk3) cells, we knocked-down IRF3 in both Efk3 and human (MRC5) cells using siRNA and infected them with MERS-CoV. The knockdown of IRF3 in MRC5 and Efk3 cells was confirmed by immune blots (Figure 5A; protein panel). Knock-down of IRF3 significantly increased the virus titer by over a hundred-fold in Efk3 cells at 48 hpi. The effect of reducing the IRF3 expression on levels of virus produced by MRC5 cells was not significant (Figure 5A). Knocking-down IRF3 significantly reduced the expression of IFNβ transcripts in virus infected Efk3 cells (Figure 5B) and increased the levels of virus produced at 48 hpi (Figure 5A). IRF3-reduced Efk3 cells also displayed a dramatic cytopathic effect (Figure 5D) compared to scrambled siRNA-treated control cells. Consistent with the IFNβ transcript expression pattern, there was a decrease in 2′,5′-oligoadenylate synthase 1 (OAS1) transcripts in IRF3 knocked-down Efk3 cells infected with MERS-CoV (Figure 5C). The effects of reducing IRF3 in MRC5 cells was not as dramatic as in bats cells (Figure 5A–D), most likely due to MERS-CoV effectively suppressing IRF3 signaling in these cells.

The siRNA-mediated IRF3 knock down in Efk3 cells was incomplete (Figure 4A) and low levels of IFNβ due to residual IRF3 in these cells may have prevented the full restoration of MERS-CoV replication in Efk3 cells to levels observed in MRC5 cells. To rule out this possibility, we infected IRF3 knockout bat cells (cr3-8; Figure 5E) with MERS-CoV along with wildtype bat (Efk3) and MRC5 cells. IRF3 deleted Efk3 cells had significantly higher virus titers 48 hpi compared to Efk3 cells, albeit still lower than MRC5 cells (Figure 5E). The virus titers at 48 and 72 hpi in IRF3 deleted bat cells were significantly higher than wildtype Efk3 cells (Figure 5F). 

## 4. Discussion

Insectivorous bats are speculated to be evolutionary hosts for MERS-CoV [8,15], an emerging coronavirus that continues to cause outbreaks in the Kingdom of Saudi Arabia. Several coronaviruses have been detected in bats in the absence of apparent disease and Jamaican fruit bats experimentally inoculated with MERS-CoV do not show overt signs of disease [10]. This suggests that, at least in some bat species, the effects of infection are not as dramatic as in humans. Munster et al. reported an interferon response in MERS-CoV infected fruit bats [10], but did not explore the mechanisms that might be involved. We have studied the molecular interactions of MERS-CoV with innate responses in bat cells to determine if and why bat cells are resilient to MERS-CoV-mediated shutdown of antiviral signaling.

To test the hypothesis that MERS-CoV infection would be modulated in bats cells because of its inability to compromise bat innate immune responses, we infected human (MRC5) and bat (Efk3) cells and studied virus replication kinetics and associated antiviral signaling. We had compared these cell lines previously [38], showing that they differed in their response to poly(I:C), a surrogate of double-stranded viral RNA. We found that as compared to MRC5 cells, the virus grew poorly in Efk3 cells and that this was likely due to continued IRF3-mediated IFNβ expression in bat cells (Figure 1A,B,F). To rule out the possibility that MRC5 and Efk3 cells were unique in their response to MERS-CoV and did not represent the human and bat response, we examined additional human lung and liver cell lines (CaLu3, A549, and Huh7), and Tb1-Lu (bat lung), the only other commercially available vespertilionid bat cell line. The virus replicated in Huh7 and CaLu3 at lower levels than in in MRC5 cells (Figure 1C). However, in these cells, viral replication was greater than in bat cells. In addition, in contrast to EfK cells, human cell lines tested did not respond with increases in IFNβ gene expression when infected with MERS-CoV. While some A549 cells displayed a limited cytopathic effect at later time points (Figure 2), MERS-CoV did not replicate to any significant extent in these cells (Figure 1C). This is consistent with other reports that A549 cells lack DPP4, the putative receptor for MERS-CoV, and are therefore not susceptible to productive viral infection [51]. Tb1-Lu cells were also not susceptible to MERS-CoV (Figure 1D). These results are consistent with those of Yang et al., who showed that Tb1-Lu cells do not express DPP4 and are only susceptible to MERS-CoV if transfected to express exogenous DPP4 [52].

Widagdo et al. mapped the distribution of DPP4, the putative receptor for MERS-CoV in the bat *Eptesicus serotinus*. Although DPP4 is present in the respiratory tract tissues of humans, in *E. serotinus*, an insectivorous bat from the same genus as *E. fuscus* (big brown bat), DPP4 was present in the kidney and intestinal tract. DPP4 was absent in the lung tissue from this bat. The authors suggest that insectivorous bats likely transmit MERS-like-CoVs via the fecal-oral route [53]. Efk3 cells express transcripts for DPP4 (Figure 1E) and we have previously shown that these cells are susceptible to MERS-CoV infection [35].

A recent study showed that DPP4 heterogeneity between hosts (human vs. bats) did not significantly affect MERS-CoV entry and propagation in baby hamster kidney (BHK) cells expressing human and different bat DPP4 receptors. In their study, Letko et al. showed that the expression of human and *Eptesicus fuscus* DPP4 in BHK cells (which are normally recalcitrant to infection) leads to almost identical levels of MERS-CoV propagation [54]. Thus, DPP4 expression levels alone may not affect virus propagation in human and bat cells. In our study, both MRC5 and Efk3 cells expressed comparable levels of transcripts for DPP4 (Figure 1E). 

MERS-CoV proteins inhibit antiviral interferon expression in human cells [19,20,23]. MERS-CoV proteins also inhibit interferon signaling in human cells [23,29], which, in turn, inhibits ISG expression and promotes virus replication. We observed an increase in IFNβ transcripts in MERS-CoV infected Efk3 cells in contrast to MRC5 cells (Figure 1F). Our observation is the first to describe the inability of MERS-CoV to suppress interferon responses in an insectivorous bat cell line. Our data support the observation by Munster et al. [10], where MERS-CoV infected Jamaican fruit bats did not display clinical signs of disease and the bats demonstrated an increase in IFNβ transcripts.

Infection with MERS- and SARS-CoVs leads to an over induction of pro-inflammatory cytokines. This “cytokine storm” causes a massive infiltration of white blood cells and an over-amplification of the inflammatory response, which in turn damages host tissue [46,55,56]. Tumor necrosis factor alpha (TNFα) is a key pro-inflammatory cytokine. We have previously shown that the synthetic double-stranded RNA surrogate, poly(I:C), does not induce a strong TNFα response in Efk3 cells, unlike in human cells [38]. We observed a similar response on infecting human (MRC5) and big brown bat cells with MERS-CoV. MRC5 cells had significantly higher levels of TNFα than Efk3 cells (Figure 1E). Big brown bats can probably limit an excessive TNFα response when infected with MERS-like-CoVs and potentially other bat CoVs. In future, ex vivo studies with bat immune cells and in vivo studies in bats will shed more light on the evolutionary adaptations that allow them to co-exist with their viruses.

IRF3 is a key transcription factor that is involved in innate antiviral signaling and antiviral interferon production in cells infected with viruses [57]. As such, several viruses have evolved mechanisms to shut down signaling through IRF3 [28,29]. We and others have shown that bat cells can produce IFNβ in response to viral infection and poly(I:C) treatment [35,58,59,60,61]. However, the role of IRF3 in the interferon signaling pathway in bats is largely unknown. In this study, we have shown that IRF3 is critical for antiviral IFNβ expression in big brown bat cells in response to poly(I:C) and MERS-CoV (Figure 4 and Figure 5). We have also shown that big brown bat IRF3 demonstrates classical signs of activation in the presence of an activation signal, i.e., poly(I:C), by translocating to the nucleus after undergoing post-translational modifications (Figure 3A,C). In Figure 3C, although cytoplasmic fractions (C) are free of nuclear contamination (lamin), the nuclear fractions (N) exhibit some cytoplasmic (calnexin) carry-over. This, however, does not affect the observation that increased amounts of a higher molecular weight band are present in the nuclear fractions of poly(I:C) treated human and bat cells. 

Multiple studies have shown that MERS-CoV has evolved multiple proteins that can effectively inhibit IRF3-mediated antiviral signaling in human cells [20,22,23,29]. Since MERS-CoV infection led to an increase in IFNβ transcripts in Efk3 cells unlike in MRC5 cells, we compared different mammalian IRF3 nucleotide sequences to determine their similarity. We observed that bat IRF3 nucleotide sequences (red box) clustered separately from human (arrow-head) and non-human primate IRF3 sequences (Figure 6). In fact, bat IRF3 sequences seem to have evolved from a common ancestor for bats and felines. Bat and feline IRF3 sequences possibly evolved from a common ancestor shared by camels (Figure 6), the likely reservoir of MERS-CoV [9,16,62]. Bats have been shown to be persistently infected with coronaviruses [63] and besides bats, cats have been shown to be persistently infected with enteric coronaviruses [64,65]. There is indeed a need to explore the role of IRF3 in persistent infections.

We noted that after poly(I:C) treatment, there was a degradation of IRF3 in MRC5 cells, unlike in big brown bat cells (Figure 4A,B; ncsiRNA and poly(I:C) treatment groups). Saitoh et al. have shown that peptidyl-prolyl cis-trans isomerase NIMA-interacting 1 (Pin1) negatively regulates IRF3 in human cells. Pin1 selectively ubiquitinates and marks phosphorylated human IRF3 for degradation, thus limiting IRF3 mediated signaling [66]. We did not observe a similar degradation of IRF3 in big brown bat cells. Preliminary data indicate that IRF3 in poly(I:C) treated Efk3 cells degrades slower than poly(I:C) treated MRC5 cells. Although we have not detected IFNα in big brown bat kidney cells, a delayed or absent negative feedback mechanism in bat cells might explain why cells from some bat species have been reported to have an unusual constitutive expression of IFNα [58], which is speculated to prime the bat antiviral responses. More work is needed to confirm the role of IRF3 in constitutive and tonic IFN expression in bats. As we build our capacity to culture immune cells in vitro from big brown bats, we will develop tools to study IRF3-Pin1 interactions in immune relevant cells. 

Knock-down of IRF3 in MERS-CoV-infected big brown bat cells decreased IFNβ transcripts with a concomitant increase in levels of MERS-CoV (Figure 5A,B), suggesting that IRF3 signaling was responsible for suppressing replication of the virus in Efk3 cells. However, there could be other signaling molecules or transcription factors in bat cells that activate IFNβ expression and more studies are required to completely dissect the IFNβ expression pathway in bats. Knocking down IRF3 in MRC5 cells did not have an effect on the levels of IFNβ transcript or MERS-CoV replication (Figure 5A,B). This is probably because MERS-CoV can effectively shut down IRF3-mediated expression of IFNs in human cells [19,20,23] and the inclusion of siRNA did not significantly further suppress it. We observed more OAS1 transcripts in cells that expressed higher levels of IFNβ transcripts. OAS1 is an ISG and its expression is largely driven by IFN expression. This data further supports our observation that IRF3 is critical for IFN and subsequent downstream gene expression in big brown bat cells.

Although there was an increase in the MERS-CoV titer in IRF3 knocked down/out bat cells as compared to wildtype bat (Efk3) cells (Figure 5A,E), IRF3 knockout bat cells did not support MERS-CoV propagation to the same extent as MRC5 cells (Figure 5E) 48 h after infection. Our data indicate that IRF3 is critical for IFNβ production in big brown bat cells (Figure 4A–C), but it is possible that these cells have alternate and yet unknown ISGs that are expressed independent of IRF3 and IFNβ expression. DeWitte-Orr et al. have shown that long double-stranded RNA can induce an antiviral response independent of IRF3 [67]. De La Cruz-Rivera et al. have shown that *Pteropus alecto* kidney cells display an atypical RNASEL induction along with a unique ISG expression profile when infected with vesicular stomatitis virus and yellow fever virus [68]. Similar adaptations may exist in insectivorous bat cells that allow them to co-exist with viruses that can otherwise inhibit these defensive responses in human cells. Further studies using siRNA-mediated knockdowns and CRISPR-Cas9-mediated knockouts of known pathways will allow us to identify if bat cells indeed have alternate and novel mechanisms to control virus replication. As more bat cells become commercially available, it will be interesting to test these cells for MERS-CoV susceptibility and downstream signaling events. It will also be interesting to test the role of tonic interferon signaling and basal levels of ISG expression in bat cells. Next generation transcriptomic studies and a better coverage of the big brown bat (*E. fuscus*) genomic sequence data will allow us to identify unique patterns of ISG regulation and expression in these cells.

Our observation that IRF3 plays a critical role in controlling MERS-CoV propagation in big brown bat cells opens up a new line of investigation. Our data suggest that although MERS-CoV can effectively suppress an antiviral IFNβ response in human cells, it is unable to subvert antiviral interferon responses in big brown bat cells. IRF3 is activated as a result of phosphorylation by cellular kinases and in the future, as more bat-specific reagents become available, it will be interesting to identify kinases in bat cells that phosphorylate bat IRF3. MERS-CoV blocks IRF3 activation by inhibiting the phosphorylation of IRF3 by cellular kinases such as TANK binding kinase 1 (TBK1) [22]. The mechanism of kinase inactivation by MERS-CoV is not yet known. Future studies will focus on identifying mechanisms that enable bat TBK1 and other cellular kinases to resist MERS-CoV protein-mediated inactivation. It is also likely that bat cells and MERS-CoV proteins have co-evolved to establish a balance between the activation and suppression of antiviral signaling. This theory will need to be tested and the ability to isolate bat CoVs may allow us to study bat CoV-IRF3 interaction in a more natural setting. This study represents yet another unique observation in cells from an insectivorous bat (*E. fuscus*) that may explain the ability of bats and their viruses to co-exist. 

Bats make up over 1200 different species and in the past, we have noticed differences in the genetic makeup between bat species [38]. It is critical to carry out these studies in cells from different bat species to identify species-specific adaptations in virus-host interactions [69]. As more knowledge about virus-host interactions in this intriguing mammalian Order become available, we may be able to adapt some of these strategies to identify new therapeutic targets or molecules to treat spill over species such as humans and agricultural animals. Emerging viruses from wildlife is a continuing problem and learning about how ancestral hosts and reservoir species co-exist with some of these viruses will help us develop prevention and treatment strategies for other species.

## Figures and Tables

**Figure 1 viruses-11-00152-f001:**
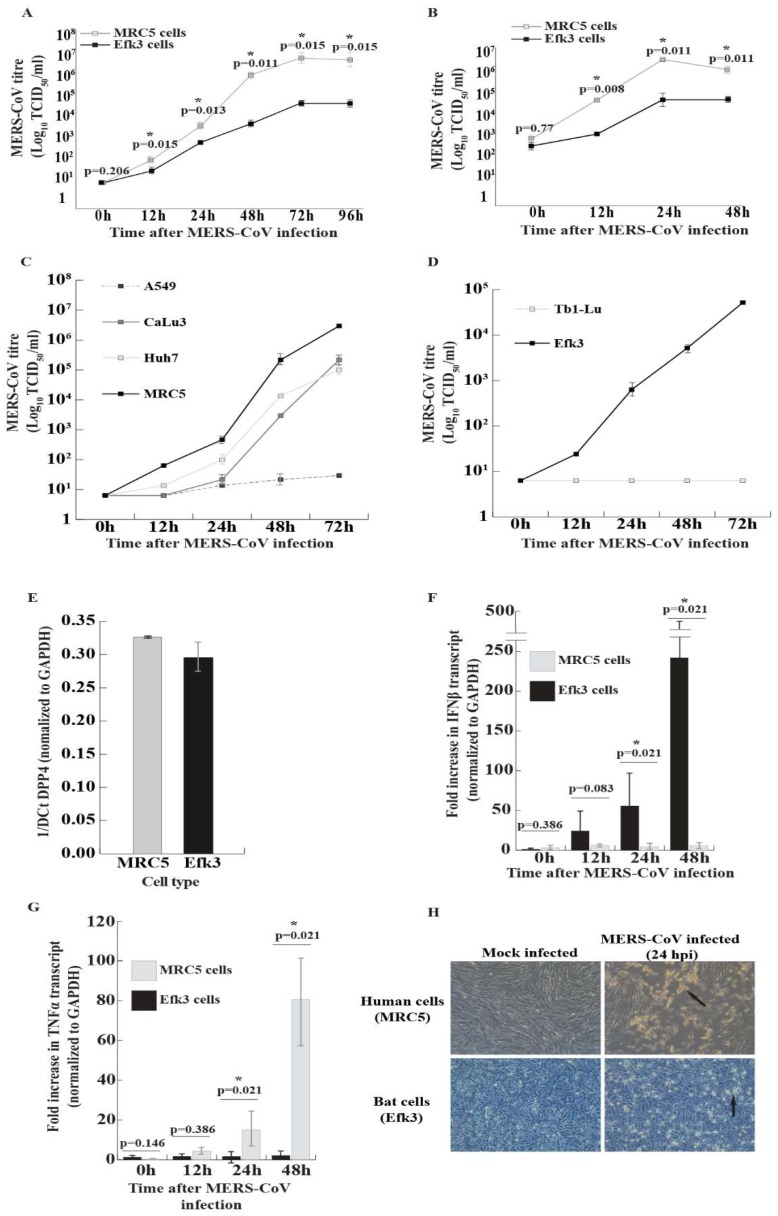
MERS-CoV replication is attenuated in bat cells and does not inhibit IFNβ responses in these cells. To assess if MERS-CoV would replicate at the same rate in human and bat cells, we infected human (MRC5, A549, CaLu3, Huh7) and bat (Efk3, Tb1-Lu) cell lines with MERS-CoV and assessed viral replication at several time-points (by TCID_50_/mL). Transcript levels for IFNβ and TNFα were quantified by qRT-PCR at the indicated time-points. (**A**) MERS-CoV replication in human (MRC5) and bat (Efk3) cells that were infected with a low multiplicity of infection (MOI) of 0.01 infectious unit/cell (mean ± SD, *n* = 3). (**B**) MERS-CoV replication in human (MRC5) and bat (Efk3) cells infected with a high MOI of 10 infectious units/cell (mean ± SD, *n* = 3). (**C**) MERS-CoV replication in human lung (A549, CaLu3 and MRC5) and liver (Huh7) cells that were infected with an MOI of 0.01 infectious units/cell (mean ± SD, *n* = 2). (**D**) MERS-CoV replication in insectivorous bat kidney (Efk3) and lung (Tb1-Lu) cells that were infected with an MOI of 0.01 infectious unit/cell (mean ± SD, *n* = 2). (**E**) Putative MERS-CoV receptor, dipeptidyl peptidase 4 (DPP4) transcript levels in MRC5 and Efk3 cells (mean ± SD, *n* = 2). (**F**) IFNβ transcript levels at different times after MERS-CoV infection in Efk3 and MRC5 cells (mean ± SD, *n* = 4). (**G**) TNFα transcript levels at several time points in MERS-CoV infected MRC5 and Efk3 cells (mean ± SD, *n* = 4). (**H**) Cytopathic effects (CPE) observed in MRC5 and Efk3 cells twenty-four hours after MERS-CoV infection (MOI = 10). qRT-PCR results are represented as fold increases over mock-infected cells, normalized to GAPDH values (see Methods). Statistical significance was calculated using the Mann Whitney *U* test for two independent samples. SD = standard deviation. *n* = number of biological replicates.

**Figure 2 viruses-11-00152-f002:**
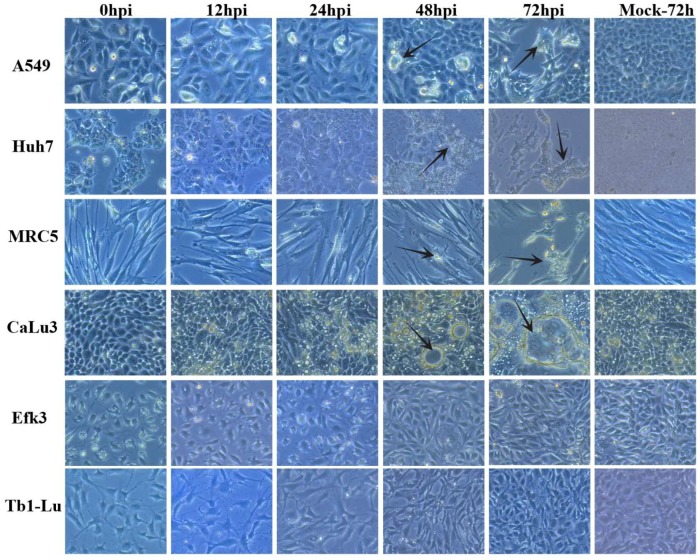
MERS-CoV causes visible cytopathic effects in human cells but not in bat cells. Cytopathic effects in human (A549, Huh7, MRC5 and CaLu3) and bat (Efk3 and Tb1-Lu) cells that were infected with MERS-CoV with an MOI of 0.01 infectious unit/cell. Arrows indicate visible cytopathic effects. hpi = hours post infection.

**Figure 3 viruses-11-00152-f003:**
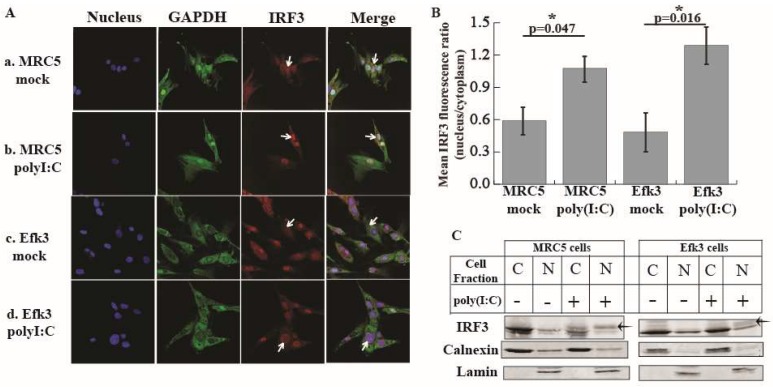
Human and bat IRF3 localize to the nucleus of the cell following poly(I:C) treatment. To determine if bat IRF3, like human IRF3, responded to poly(I:C)-mediated activation by post-translational modification and nuclear translocation, we performed immunofluorescent microscopy and immunoblots on poly(I:C)-treated and mock-treated cells. (**A**) The cellular location of endogenous IRF3 in mock and poly(I:C) treated human (MRC5) and bat (Efk3) cells. IRF3 is stained red. GAPDH is stained green to highlight the cellular cytoplasm and the nucleus is stained blue. (**B**) Mean IRF3 fluorescence ratio (nucleus:cytoplasm) in MRC5 and Efk3 cells (mean ± SD, *n* = 5). (**C**) Immune blots of nuclear and cytoplasmic fractions of mock and poly(I:C) treated MRC5 and Efk3 cells. C = cytoplasmic fraction, N = nuclear fraction, arrow = higher molecular weight IRF3, calnexin = cytoplasmic marker, and lamin = nuclear marker. Statistical significance was calculated using the Mann Whitney *U* test for two independent samples. * *p* < 0.05. SD = standard deviation. *n* = number of fields. For the original, full size blots see Appendix A.

**Figure 4 viruses-11-00152-f004:**
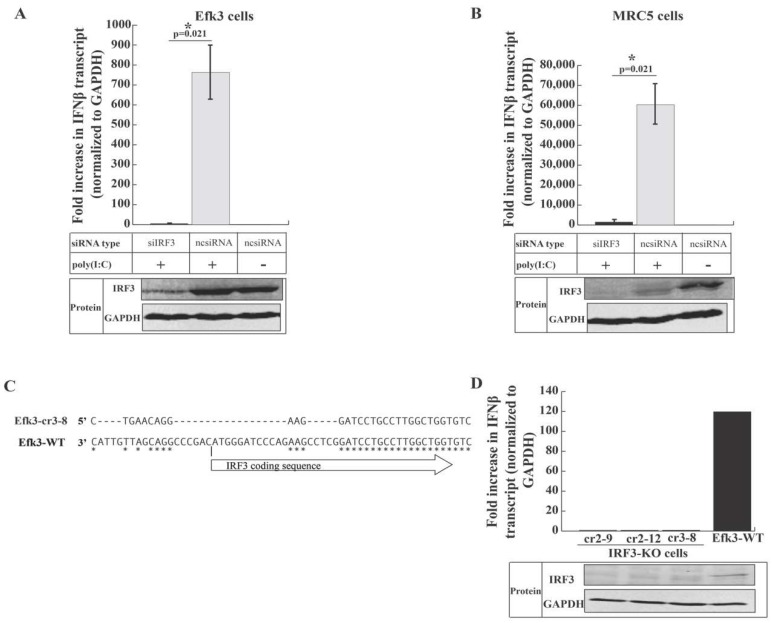
IRF3 is required for IFNβ signaling in response to poly(I:C) in human and bat cells. To determine the role of IRF3 in antiviral interferon signaling, we partially knocked-down IRF3 in both bat (Efk3) and human (MRC5) cells and quantified the increase in IFNβ transcripts in these cells in response to poly(I:C) by quantitative real-time PCR (qRT-PCR). To further demonstrate the dependency of poly(I:C)-mediated expression of IFNβ on IRF3 in Efk3 cells, we generated IRF3 knockout bat cells and quantified the increase in IFNβ transcripts in these cells in response to poly(I:C) by quantitative real-time PCR (qRT-PCR). (**A**) Fold increase in IFNβ transcript levels in IRF3 knocked-down Efk3 cells (siIRF3) and negative control siRNA (ncsiRNA) treated cells on stimulation with poly(I:C) (mean ± SD, *n* = 4). (**B**) Fold increase in IFNβ transcript levels in IRF3 knocked-down MRC5 cells (siIRF3) and negative control siRNA (ncsiRNA) treated cells on stimulation with poly(I:C) (mean ± SD, *n* = 4). (**C**) To generate IRF3 knockout Efk3 cells, we deleted a portion of the first exon in the genomic DNA using CRISPR-Cas9. This schematic represents the deletion in the first exon of IRF3 in Efk3-c3-8 cells, one of the IRF3 deleted cell lines. “ATG” in the wildtype cells (Efk3-WT) marks the start codon of IRF3. Similar nucleotides are indicated by “*”. (**D**) Fold increase in IFNβ transcript levels in IRF3 knocked-out bat cells (IRF3-KO cells) and wildtype bat cells (Efk3-WT). IRF3 knockdown and knockout were confirmed by immune blots (protein panel; Figures A,B,D). Results are represented as fold increases over mock poly(I:C) transfected cells, normalized to GAPDH values (see Methods). Statistical significance was calculated using the Mann Whitney *U* test for two independent samples. * *p* < 0.05. SD = standard deviation. *n* = number of biological replicates. For full size blots, see Appendix A.

**Figure 5 viruses-11-00152-f005:**
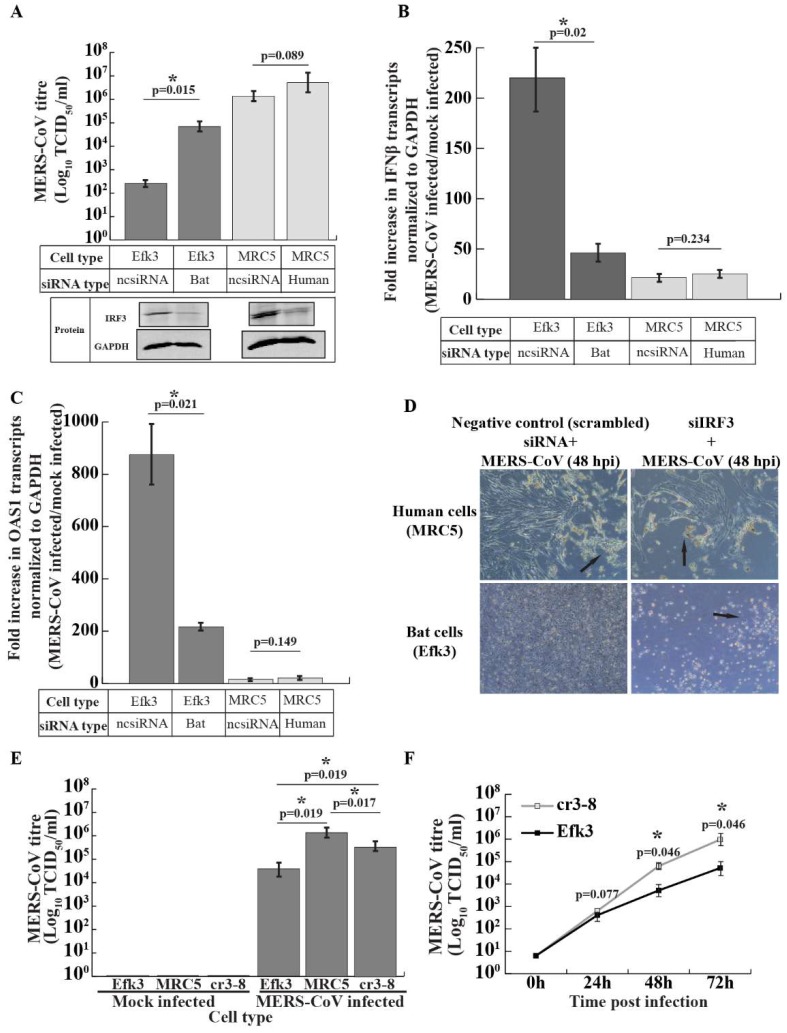
Reduction (knock-down) or deletion (knock-out) of IRF3 increases levels of MERS-CoV and decreases IFNβ transcripts in big brown bat cells. To determine if IRF3 mediated signaling is crucial for limiting MERS-CoV replication in big brown bat cells, we knocked-down IRF3 in bat (Efk3) and human (MRC5) cells and infected them with MERS-CoV. A multiplicity of infection (MOI) of 0.01 or 0.1 infectious unit/cell was used for virus propagation studies and an MOI of 10 infectious units/cell was used for cytokine studies. We quantified virus replication in these cells by the TCID_50_ assay 48 h post infection (hpi) and transcripts for IFNβ and OAS1 by qRT-PCR 24 hpi. (**A**) MERS-CoV titers in MRC5 and Efk3 cells that were either treated with human or bat siIRF3 (IRF3 knockdown), respectively, or ncsiRNA (mock/no IRF3 knockdown) 48 hpi (mean ± SD, *n* = 4). IRF3 knockdown in MRC5 and Efk3 cells by siRNA was confirmed by immune blots (protein panel). An MOI of 0.01 infectious unit/cell was used for MERS-CoV infections. (**B**) Fold change in IFNβ transcript levels in siIRF3 (IRF3 knockdown) or ncsiRNA (mock/no IRF3 knockdown) treated Efk3 and MRC5 cells infected with MERS-CoV (mean ± SD, *n* = 4). An MOI of 10 infectious units/cell was used for the infections. (**C**) Fold change in interferon stimulated gene (OAS1) transcript levels in siIRF3 (IRF3 knockdown) or ncsiRNA (mock/no IRF3 knockdown) treated Efk3 and MRC5 cells infected with MERS-CoV (mean ± SD, *n* = 4). An MOI of 10 infectious units/cell was used for the infections. (**D**) Cytopathic effect (CPE) observed in siIRF3 (IRF3 knockdown) or ncsiRNA (mock/no IRF3 knockdown) treated Efk3 and MRC5 cells 48 hpi with MERS-CoV. (**E**) MERS-CoV titer in bat (Efk3), human (MRC5) and IRF3 knockout bat (cr3-8) cells 48 hpi with an MOI of 0.1 infectious unit/cell (mean ± SD, *n* = 4). (**F**) MERS-CoV titer in wildtype bat (Efk3) and IRF3 deleted bat (cr3-8) cells infected with an MOI of 0.01 infectious unit/cell over a period of 72 h. qRT-PCR results are represented as fold increases over mock infected cells, normalized to GAPDH values (see Methods). Statistical significance was calculated using the Mann Whitney *U* test for two independent samples. * *p* < 0.05. SD = standard deviation. *n* = number of biological replicates. For full size blots, see Appendix A.

**Figure 6 viruses-11-00152-f006:**
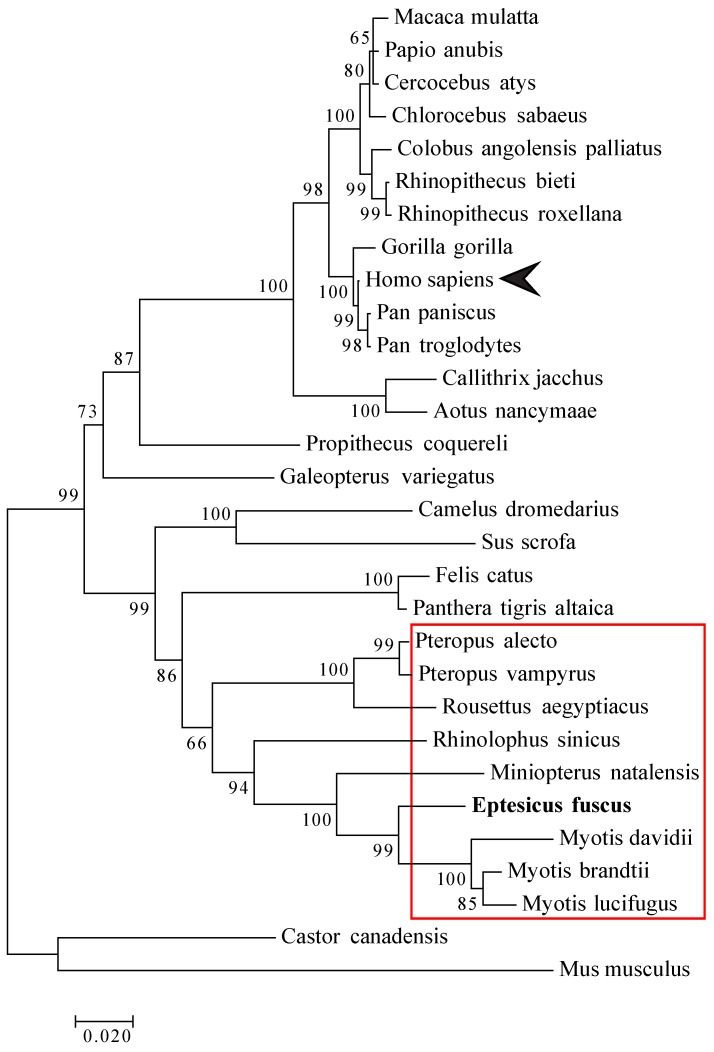
Bat IRF3 nucleotide sequences are divergent from their human counterpart. IRF3 nucleotide sequences for several other mammals and big brown bat IRF3 sequence (bold) were aligned to identify similarities. The maximum likelihood tree for IRF3 nucleotide sequences (1000 Bootstrap) is represented here. Bar represents nucleotide substitution per site. For IRF3 nucleotide sequence information, see Table 2.

**Table 1 viruses-11-00152-t001:** Primer and siRNA sequences.

Name	Sequence-human (5′-3′)	Sequence-*E. fuscus* (5′-3′)	Feature
IFNβ	GCTTGGATTCCTACAAAGAAGCA	GCTCCGATTCCGACAGAGAAGCA	Forward primer
	ATAGATGGTCAATGCGGCGTC	ATGCATGACCACCATGGCTTC	Reverse primer
GAPDH	GGAGCGAGATCCCTCCAAAAT	GGAGCGAGATCCCGCCAACAT	Forward primer
	GGCTGTTGTCATACTTCTCATGG	GGGAGTTGTCATACTTGTCATGG	Reverse primer
TNF	CAGCCTCTTCTCCTTCCTGA	GCCCATGTTGTAGCAAACC	Forward primer
	AGATGATCTGACTGCCTGGG	GCCCTTGAAGAGGACCTGGG	Reverse primer
OAS1	AGTTGACTGGCGGCTATAAAC	AGGTGACGGACGACTACAGAC	Forward primer
	GTGCTTGACTAGGCGGATGAG	GTGCTTGACCAGGCGGATGAG	Reverse primer
DPP4	TGACATGGGCAACACAAGA	TGATCTTGCCTCCTCATTTTGATAA	Forward primer
	AACCCAGCCAGTAGTACTC	GTAACCACTTCCTCTGCCATCAA	Reverse primer
siIRF3-1 (Duplex)	5′ rGrUrGrGrArGrGrCrArGrUrArCrUrUrCrUrGrArUrArCrCCA 3′	5′ rCrArArGrArArGrCrUrArGrUrGrArUrGrGrUrCrArArGrGTT 3′	r=Ribose sugar
	5′ rUrGrGrGrUrArUrCrArGrArArGrUrArCrUrGrCrCrUrCrCrArCrCrA 3′	5′ rArArCrCrUrUrGrArCrCrArUrCrArCrUrArGrCrUrUrCrUrUrGrGrU 3′	r=Ribose sugar
siIRF3-2 (Duplex)	5′ rArCrUrGrUrGrGrArCrCrUrGrCrArCrArUrUrUrCrCrArACA 3′	5′ rCrUrGrCrCrArArCrCrUrGrGrArArGrArGrGrArArUrUrUCA 3′	r=Ribose sugar
	5′ rUrGrUrUrGrGrArArArUrGrUrGrCrArGrGrUrCrCrArCrArGrUrArU 3′	5′ rUrGrArArArUrUrCrCrUrCrUrUrCrCrArGrGrUrUrGrGrCrArGrGrU 3′	r=Ribose sugar

**Table 2 viruses-11-00152-t002:** Accession numbers for mammalian IRF3 nucleotide sequences.

Species Name	Accession Number
*Aotus nancymaae*	XM_012460874
*Callithrix jacchus*	XM_002762377
*Camelus dromedarius*	XM_010993178
*Castor Canadensis*	XM_020159968
*Cercocebus atys*	NM_001305969 XM_012080898
*Chlorocebus sabeaus*	XM_007997589
*Colobus angolensis palliatus*	XM_011946361
*Eptesicus fuscus*	XM_008154348
*Felis catus*	XM_003997503
*Galeopterus variegatus*	XM_008575391
*Gorilla gorilla*	XM_019015447
*Homo sapiens*	NM_001571
*Macaca mulatta*	NM_001135797 XM_001115379
*Miniopterus natalensis*	XM_016206049
*Mus musculus*	NM_016849
*Myotis brandtii*	XM_014550465
*Myotis davidii*	KU161111
*Myotis lucifugus*	XM_014449832
*Pan paniscus*	XM_003814337
*Pan troglodytes*	XM_016936566
*Panthera tigris altaica*	XM_007074177
*Papio anubis*	XM_009194993
*Propithecus coquereli*	XM_012643169
*Pteropus alecto*	XM_006905022
*Pteropus vampyrus*	XM_011374528
*Rhinolophus sinicus*	XM_019741528
*Rhinopithecus bieti*	XM_017882166
*Rhinopithecus roxellana*	XM_010369491
*Rousettus aegyptiacus*	XM_016122379
*Sus scrofa*	NM_213770

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
