# Peer review of "Interferon Regulatory Factor 3-Mediated Signaling Limits Middle-East Respiratory Syndrome (MERS) Coronavirus Propagation in Cells from an Insectivorous Bat"

_viruses, 2019, doi:10.3390/v11020152_

Round 1
Reviewer 1 Report
The article ‘Interferon response factor 3-mediated signaling limits Middle-East respiratory syndrome (MERS) coronavirus propagation in cells from an insectivorous bat’ investigated on MERS-CoV infection in Eptesicus fuscus (big brown bat) kidney (Efk3) and to study the immunology of Efk3 governing the susceptibility of MERS-CoV.
MERS-CoV propagated to significantly lower titers in big brown bat cells when it was compared to the level of human cells.The DPP4 expression of the cell lines were investigated and both human lung (MRC5) and Efk3 cells had comparable levels of DPP4 transcripts, suggesting that receptor level is not the determining factor.
The authors hypothesized that the immunology of Efk3 cells played a role for the lower MERS-CoV replication. A significant increase in IFNb transcripts at later time points of 24 and 48 hpi in Efk3 cells were induced by MERS-CoV infection comparing to Huh7 cells. Lower levels of TNFa transcripts were induced in Efk3 cells as well.
The authors thus went further upstream into interferon regulatory factor 3 (IRF3), which mediates the antiviral interferon signaling. IRF3 was detected localizing in the nucleus of big brown bat cells in response to poly (I:C), which is a hallmark of IRF3 activation and it is present in bat cells.
The authors then make use of knockdown/knockout approaches to observe for expression changes in the bat cell immunology. When IRF3 was knocked down in both MRC5 and Efk3 cells, the expression of IFNb transcripts in response to poly (I:C) was significantly reduced in both cells. IRF3 knockout bat cells by CRISPR-Cas9 did not respond to poly (I:C) stimulation with no IFNb transcripts relative to wildtype bat Efk3 cells.
Knock-down of IRF3 significantly increased MERS-CoV titre by over a hundred fold in Efk3 cells at 48 hpi. The effect of reducing IRF3 expression on levels of virus produced by MRC5 cells was however not significant. Knocking-down IRF3 significantly reduced the expression of IFNb transcripts and 2’, 5’- oligoadenylate synthase 1 (OAS1) in virus infected Efk3 cells and increased virus produced at 48 hpi to levels similar to virus-infected MRC5 cells. IRF3-reduced Efk3 cells also displayed dramatic cytopathic effect. The effects of reducing IRF3 in MRC5 cells was not as dramatic as in bats cells, most likely due to MERS-CoV effectively suppressing IRF3 signaling in these cells.
The message of this article is that the continued IRF3-mediated IFNb expression in bat cells as well as the inability of MERS-CoV to suppress interferon responses in an insectivorous bat cell line contributed to the outcome of MERS infection in Efk3 cells. The author suggested that Bat IRF3 nucleotide sequences clustered separately from human and non-human primate IRF3 sequences phylogenetically, which MERS-CoV could inhibit them in different efficiency. The study is organized and well written which could bring insight to the field of virology and bat zoonosis.
1. Minor comment:
Fold increase with respect to a reference would be a clearer presentation in y-axis comparing to 1/DCt. It would be more consistent for the whole paper to use 1 way of presentation.
2. Figure 4. It seems that the drop of IFN beta level for Efk3 cells and MRC5 cells after IRF3 siRNA knockdown were in different magnitude (Efk3 cells: 750 fold vs MRC5: 60000 fold) while the baseline level of IFN beta in Efk3 cells were actually higher than MRC5 cells. The fold change should be stated quantitatively instead of ‘significantly reduced in both cells’. Is such fold change in reduction having implication of the differential efficiency of IRF3 signaling in both type of cells?
Author Response
We would like to thanks the reviewer for their very helpful and thorough review. We appreciate the time this valuable service takes.
Reviewer's comments and our response (in italics):
The article ‘Interferon response factor 3-mediated signaling limits Middle-East respiratory syndrome (MERS) coronavirus propagation in cells from an
insectivorous bat’ investigated on MERS-CoV infection in Eptesicus fuscus (big brown bat) kidney (Efk3) and to study the immunology of Efk3 governing the susceptibility of MERS-CoV.
MERS-CoV propagated to significantly lower titers in big brown bat cells when it was compared to the level of human cells.The DPP4 expression of the cell
lines were investigated and both human lung (MRC5) and Efk3 cells had comparable levels of DPP4 transcripts, suggesting that receptor level is not the
determining factor.
The authors hypothesized that the immunology of Efk3 cells played a role for the lower MERS-CoV replication. A significant increase in IFNb transcripts at
later time points of 24 and 48 hpi in Efk3 cells were induced by MERS-CoV infection comparing to Huh7 cells. Lower levels of TNFa transcripts were
induced in Efk3 cells as well.
The authors thus went further upstream into interferon regulatory factor 3 (IRF3), which mediates the antiviral interferon signaling. IRF3 was detected
localizing in the nucleus of big brown bat cells in response to poly (I:C), which is a hallmark of IRF3 activation and it is present in bat cells.
The authors then make use of knockdown/knockout approaches to observe for expression changes in the bat cell immunology. When IRF3 was knocked
down in both MRC5 and Efk3 cells, the expression of IFNb transcripts in response to poly (I:C) was significantly reduced in both cells. IRF3 knockout
bat cells by CRISPR-Cas9 did not respond to poly (I:C) stimulation with no IFNb transcripts relative to wildtype bat Efk3 cells.
Knock-down of IRF3 significantly increased MERS-CoV titre by over a hundred fold in Efk3 cells at 48 hpi. The effect of reducing IRF3 expression on levels of virus produced by MRC5 cells was however not significant. Knocking-down IRF3 significantly reduced the expression of IFNb transcripts and 2’, 5’-
oligoadenylate synthase 1 (OAS1) in virus infected Efk3 cells and increased virus produced at 48 hpi to levels similar to virus-infected MRC5 cells. IRF3-
reduced Efk3 cells also displayed dramatic cytopathic effect. The effects of reducing IRF3 in MRC5 cells was not as dramatic as in bats cells, most likely
due to MERS-CoV effectively suppressing IRF3 signaling in these cells.
The message of this article is that the continued IRF3-mediated IFNb expression in bat cells as well as the inability of MERS-CoV to suppress
interferon responses in an insectivorous bat cell line contributed to the outcome of MERS infection in Efk3 cells. The author suggested that Bat IRF3
nucleotide sequences clustered separately from human and non-human primate IRF3 sequences phylogenetically, which MERS-CoV could inhibit them
in different efficiency. The study is organized and well written which could bring insight to the field of virology and bat zoonosis.
1. Minor comment:
Fold increase with respect to a reference would be a clearer presentation in yaxis comparing to 1/DCt. It would be more consistent for the whole paper to
use 1 way of presentation.
Authors’ response: We have removed figure 4C, which was a representation of data in Figs 4A and 4B as dCT values. The original figures 4D and E are now 4C and D respectively.
As suggested by the reviewer, we have also replaced figures 5B and 5C to now include plots with fold change values. In the new plots, we have plotted fold change in IFNband OAS1 transcript levels between mock infected and MERS-CoV infected cells, normalized to GAPDH.
2. Figure 4. It seems that the drop of IFN beta level for Efk3 cells and MRC5
cells after IRF3 siRNA knockdown were in different magnitude (Efk3 cells: 750
fold vs MRC5: 60000 fold) while the baseline level of IFN beta in Efk3 cells
were actually higher than MRC5 cells. The fold change should be stated
quantitatively instead of ‘significantly reduced in both cells’.
Authors’ response: We have mentioned the quantitative values in the text as suggested by the reviewer.
Although bat and human cells treated with control siRNA (ncsiRNA) have different expression levels for Interferon beta when stimulated with polyI:C, there is a significant reduction in interferon beta levels in both cell types treated with IRF3 specific siRNA (siIRF3). The baseline levels of Interferon beta in both cell types are similar (negligible) in these plots (Figs 4A and 4Bs; ncsiRNA and polyI:C ‘-‘ bars).
Is such fold change in reduction having implication of the differential efficiency of IRF3 signaling in both type of cells?
Authors’ response: The observed differences can be attributed to several factors including the transfection efficiencies of siRNA and poly(I:C) in these cells. However, the data demonstrates that IRF3 is primarily responsible for double-stranded RNA (polyIC)-mediated interferon expression in both human (MRC5) and bat (Efk3) cells.
Reviewer 2 Report
In the current manuscript titled “Interferon response factor 3-mediated signaling limits Middle East respiratory syndrome (MERS)-coronavirus propagation in cells from an insectivorous bat” Banerjee A et al., investigate the mechanistic basis for the persistence of MERS-CoV in bats, the reservoirs of multiple human and animal viruses. Infecting human lung and bat kidney cell lines with MERS-CoV, the authors test the hypothesis that the inability of MERS-CoV to suppress bat anti-viral interferon response, in part contributes to strong IFN and interferon-stimulated gene responses thus limiting MERS-CoV replication. In contrast, the authors find that MERS-CoV replicates to high titers in human lung epithelial MRC5 cells by efficiently antagonizing anti-viral IFN/ISG responses. Further, employing siRNA knockdown and CRISPR-CAS9 knockout studies; the authors show that IRF3 signaling is essential for IFN-beta expression in bat but not human epithelial cells.
The results demonstrated in this study are important and likely explain viral persistence in the bat population as compared to other mammalian hosts. The experiments, to a large extent, are well designed, and the manuscript is well written. However, there are several caveats that need to be addressed to strengthen this paper and justify the author's conclusions. The comments are listed below.
Major Comments:
1) Recent studies suggest that the constitutive expression of antiviral genes, particularly IFN-alpha, STAT-1 and interferon-stimulated genes (Zhou P et al., PNAS 2016 and Schountz T et al., 2017) likely contribute to controlled virus replication and low level of shedding in bat cells. These studies also show that IFN-beta level is low in bats, and can be induced upon stimulation/virus challenge in bat cells. The current study proposes that the inability of MERS-CoV to antagonize antiviral responses in bat cells results in robust IFN/ISG response, which in-turn reduces virus replication in bat cells. To conclusively establish these claims, the authors need to demonstrate the levels of IFN-alpha in bat and human cells infected with MERS-CoV. This is important as the authors use only IFN-beta as the primary readout of IFN response throughout the study. It is also possible that by knocking out/knocking down IRF3, the authors are abrogating constitutive IFN-alpha/ISG expression, resulting in increased virus replication as shown in Figure 5. This is particularly important given siRNA knockdown caused an only ~2-fold reduction in IFN-beta/OAS1 expression and a 100 fold increase in MERS-CoV replication (Figure 5A-C). In addition, please discuss above mentioned and related studies in current results.
2) Figure 2: The authors show no MERS-CoV replication in A549 cells (Figure 1C). However, figure 2 shows CPE at 48 and 72 hpi in these cells. Please explain the reason for this discrepancy.
3) Figure 3 and 4: The authors demonstrate IRF-3/pIRF3 expression in bat and human cells upon poly-I:C challenge. However, there is no data to show IRF3 expression after MERS-CoV infection. Figure 5A does show IRF3 expression in MERS-CoV infected cells, but it is difficult to assess as presented since there is no mock infected IRF3 control.
4) Figure 3: The antibody author’s used in this study to demonstrate nuclear localization of IRF3 is reactive in humans and mice (Abcam 68481). Did the authors use the same antibody to show bat IRF3? If so, is it cross-reactive in bats? Is the peptide used to generate this antibody has a similar sequence to that of bats?
5) Figure 4 and 5: It is essential to include no poly I:C transfection/no MERS-CoV infection si-IRF3 controls to demonstrate the basal levels of IFNa/b in the absence of IRF3. Also, provide no infection si-IRF3 control for figure 5D. Showing the level of IFN-a/b levels in CRISPR-cas9 IRF3 knock out cells will significantly strengthen this figure.
Minor Comments:
1) Line 257: provide the reference of Figure 1A-1C, instead of just 1C.
2) Line 276: The refer huh7 cells as kidney cells. The huh7 cells are liver carcinoma-derived cells. This requires correction.
3) Line 383-384: The authors state that IRF3 knockdown increases virus titers in bat cells to match human cells. This statement is incorrect as virus titers are still ~1log lower bat IRF3 knockdown cells compared to MRC5 cells.
4) Line 391-392: Correct this sentence.
5) Figure 5A and 5E: MERS-CoV titer in control efk3 bat cells of 5A is ~102 and in 5E is >104. Provide an explaination for the discrepancy.
6) Line 512-514; Authors refer that Pin1 negatively regulates IRF3 levels. Include a reference to show that Pin1 is unregulated upon PI:C treatment or virus infection.
7) Line 526-528: Include reference/s to support this statement.
Author Response
We would like to thanks the reviewer for their very helpful and thorough review. We appreciate the time this valuable service takes.
Reviewer's comments and our response (in italics):
In the current manuscript titled “Interferon response factor 3-mediated signaling limits Middle East respiratory syndrome (MERS)-coronavirus propagation in cells from an insectivorous bat” Banerjee A et al., investigate the mechanistic basis for the persistence of MERS-CoV in bats, the reservoirs of multiple human and animal viruses. Infecting human lung and bat kidney cell lines with MERS-CoV, the authors test the hypothesis that the inability of MERS-CoV to suppress bat anti-viral interferon response, in part contributes to strong IFN and interferon-stimulated gene responses thus limiting MERS-CoV replication. In contrast, the authors find that MERS-CoV replicates to high titers in human lung epithelial MRC5 cells by efficiently antagonizing anti-viral IFN/ISG responses. Further, employing siRNA knockdown and CRISPR-CAS9 knockout studies; the authors show that IRF3 signaling is essential for IFN-beta expression in bat but not human epithelial cells.
The results demonstrated in this study are important and likely explain viral persistence in the bat population as compared to other mammalian hosts. The experiments, to a large extent, are well designed, and the manuscript is well written. However, there are several caveats that need to be addressed to strengthen this paper and justify the author's conclusions. The comments are listed below.
Major Comments:
1) Recent studies suggest that the constitutive expression of antiviral genes, particularly IFN-alpha, STAT-1 and interferon-stimulated genes (Zhou P et al., PNAS 2016 and Schountz T et al., 2017) likely contribute to controlled virus replication and low level of shedding in bat cells. These studies also show that IFN-beta level is low in bats, and can be induced upon stimulation/virus challenge in bat cells. The current study proposes that the inability of MERS-CoV to antagonize antiviral responses in bat cells results in robust IFN/ISG response, which in-turn reduces virus replication in bat cells. To conclusively establish these claims, the authors need to demonstrate the levels of IFN-alpha in bat and human cells infected with MERS-CoV. This is important as the authors use only IFN-beta as the primary readout of IFN response throughout the study. It is also possible that by knocking out/knocking down IRF3, the authors are abrogating constitutive IFN-alpha/ISG expression, resulting in increased virus replication as shown in Figure 5. This is particularly important given siRNA knockdown caused an only ~2-fold reduction in IFN-beta/OAS1 expression and a 100 fold increase in MERS-CoV replication (Figure 5A-C). In addition, please discuss above mentioned and related studies in current results.
Authors’ response: We have previously assayed our cells for Interferon alpha expression in response to poly(I:C), CpG ODN and single-stranded RNA. Big brown bat kidney (Efk3) cells do not respond in terms of interferon alpha expression [1]. Interferon alpha is primarily secreted by immune cells and we are hoping to assay for interferon alpha in bone marrow derived myeloid cells. Furthermore, the observation of constitutive interferon expression may be species specific as recently reported by Pavlovich et al. [2], where the authors did not detect constitutive interferon expression in primary cells from Rousettus bats. In our opinion, constitutive/basal interferon expression levels may also depend on the cell type and the immortalization procedure employed to establish the cell lines (reviewed in MDPI Viruses, manuscript ID: 421549; https://www.mdpi.com/1999-4915/11/1/41/htm).
Although we do not observe higher basal levels of interferon beta expression in our bat cells, the idea of tonic interferon signaling is interesting. As mentioned by the reviewer, disrupting IRF3 may also affect constitutive ISG expression (tonic IFN signaling). This further highlights the importance of IRF3-mediated signaling in bat cells that allow them to effectively control MERS-CoV replication. We have added this point in the discussion section of the manuscript.
The y-axis of the OAS1 graph seemed confusing and we have now replaced it with fold changes (as suggested by reviewer 1).
We have discussed Zhou et al.’s article in the discussion section of our manuscript (lines 537-540).
2) Figure 2: The authors show no MERS-CoV replication in A549 cells (Figure 1C). However, figure 2 shows CPE at 48 and 72 hpi in these cells. Please explain the reason for this discrepancy.
Authors’ response: The increase in MERS-CoV titres in A549 cells was very limited. Although we could see a small increase in virus titre at 48 and 72 hours post infection, this was very little compared to other cell lines (Figure 1C). In figure 2, we have highlighted the limited foci of virus replication in A549 cells at later time points of 48 and 72 hours post infection. We observed one to two foci of replication only, in each infected well. We could have included a field of no virus replication, but these series of images will allow other researchers to identify zones of MERS-CoV replication (CPE) in different cell lines. Such images are largely lacking for a wide variety of cells that have been used to study MERS-CoV. We have also discussed this observation in the discussion section (Lines 469-471).
3) Figure 3 and 4: The authors demonstrate IRF-3/pIRF3 expression in bat and human cells upon poly-I:C challenge. However, there is no data to show IRF3 expression after MERS-CoV infection. Figure 5A does show IRF3 expression in MERS-CoV infected cells, but it is difficult to assess as presented since there is no mock infected IRF3 control.
Authors’ response: We wanted to study IRF3 expression and localization in bat and human cells by microscopy after MERS-CoV infection, but the mandated step to fix the cells with either 10% Neutral buffered formalin or paraformaldehyde skewed our data. Methanol alone has not been tested for MERS-CoV inactivation and as such, we could not take our samples out of containment after methanol fixation. We recently published this discrepancy in IRF3 cellular localization based on the choice of fixative used to fix the cells [3]. In a separate study, we are looking at understanding the negative regulation (and differences in the kinetics) of IRF3 in bat cells by Pin1. We will elaborate on the IRF3 feedback story in a future manuscript.
4) Figure 3: The antibody author’s used in this study to demonstrate nuclear localization of IRF3 is reactive in humans and mice (Abcam 68481). Did the authors use the same antibody to show bat IRF3? If so, is it cross-reactive in bats? Is the peptide used to generate this antibody has a similar sequence to that of bats?
Authors’ response: Yes. We tested this antibody for cross-reactivity with bat and human IRF3. This antibody was raised against a conserved region of IRF3 protein. This region is conserved in big brown bat IRF3 amino acid sequence as well. We have previously published with this antibody where we have detected both bat and human IRF3 proteins [3].
5) Figure 4 and 5: It is essential to include no poly I:C transfection/no MERS-CoV infection si-IRF3 controls to demonstrate the basal levels of IFNa/b in the absence of IRF3. Also, provide no infection si-IRF3 control for figure 5D. Showing the level of IFN-a/b levels in CRISPR-cas9 IRF3 knock out cells will significantly strengthen this figure.
Authors’ response: In figures 4 and 5, mock (no poly I:C transfection/no MERS-CoV) Ct values have been used to plot DDCt (fold change) values for the respective test samples.
There was no difference in basal levels of IFN beta in mock Efk3 (bat) and MRC5 (human) cells. This has been recently reported by Pavlovich et al. where unlike P. alecto cells, cells from the bat Rousettus aegyptiacus do not demonstrate higher levels of basal interferon expression [2]. Thus, differences in basal levels of interferon beta may exist at the cellular and species level for bats. These differences may also be artifacts of the immortalization procedure used to generate cell lines form bats. We have discussed this in a previously published article [4]and a recently accepted article (MDPI Viruses manuscript ID: 421549).
Since we have shown that IRF3 knockout cells do not express IFN-b after poly(I:C) treatment (figure 4E), we thought it was unnecessary to repeat the observation with a virus infection. Furthermore, we have also demonstrated that knocking down IRF3 using siRNA reduces both poly(I:C) (Figures 4A and 4B) and MERS-CoV-mediated expression (Figure 5B) of interferon beta.
The objective of Figure 5D was to demonstrate the difference in CPE observed in wildtype or IRF3-reduced bat and human cells that are infected with MERS-CoV.Mock infected and no siRNA transfected cells are shown in Figure 1H (mock infected panel).
Minor Comments:
1) Line 257: provide the reference of Figure 1A-1C, instead of just 1C.
Authors’ response: done.
2) Line 276: The refer huh7 cells as kidney cells. The huh7 cells are liver carcinoma-derived cells. This requires correction.
Authors’ response: done.
3) Line 383-384: The authors state that IRF3 knockdown increases virus titers in bat cells to match human cells. This statement is incorrect as virus titers are still ~1log lower bat IRF3 knockdown cells compared to MRC5 cells.
Authors’ response: We have removed the phrase ‘match human cells’.
4) Line 391-392: Correct this sentence.
Authors’ response: In our opinion, this sentence adequately and accurately describes the results.
5) Figure 5A and 5E: MERS-CoV titer in control efk3 bat cells of 5A is >102 and in 5E is >104. Provide an explaination for the discrepancy.
Authors’ response: The difference is likely due to the amount of input virus in these experiments. In figure 5A, an MOI of 0.01infectious unit/cell was used (previously mentioned in the figure legend). In Fig 5E, an MOI of 0.1 infectious unit/cell was used (also mentioned in the figure legend).
6) Line 512-514; Authors refer that Pin1 negatively regulates IRF3 levels. Include a reference to show that Pin1 is unregulated upon PI:C treatment or virus infection.
Authors’ response: To our knowledge, reference 66 (Saitoh et al.) is the only reference that has studied the negative regulation of IRF3 by Pin1. The authors used double-stranded RNA (polyI:C) in their studies.
7) Line 526-528: Include reference/s to support this statement.
Authors’ response: done.
Reviewer 3 Report
In this work by Banerjee et. al. examine the antiviral activity of Middle East Respiratory syndrome (MERS) coronavirus (CoV) infection in bat cells. Here, the authors demonstrate that MERS-CoV grew to slightly higher titers in human cells which is related to their capacity to suppress interferon (IFN) expression albeit to a lesser extent in bat cells. Through this work, they demonstrate that IRF3 plays a critical role in mediating IFN response during MERS-CoV infection.
The manuscript examines an important aspect of MERS-CoV infection and is well written but the major critique of this manuscript is that authors only observe modest effect with MERS-CoV growth kinetics and IFN responses in human vs bat cells and authors fail to decipher what is the major contributing factor, for example why they see less suppression of IRF3 in bat cells in comparison to human cells. Thus, a role for bat IRF3 in host response to MERS-CoV infection or why it may be differs from human remains unclear. In addition, strong data supporting the conclusion in the abstract that bat cells are resistant to MERS-CoV mediated interferon responses and the biological relevance of this function has not been provided.
Some major concerns:
The authors only examine the effect of IRF3 butwhat about the effect on IFN signaling pathway, such as effect on IFN signaling via the JAK/STAT pathway in the two cell lines. What is effect on expression of IFN stimulated genes.
Authors used different cell lines but not cell line that are defective in IFN production such as Vero cells, will the virus grow better in Vero cells as compared to IFN competent cells?
In Fig 3, the effect on IRF3 localization is hard to interpret. As the authors have the known coding sequence for bat IRF3, it would be cleaner experiment if authors clone bat IRF3 to generate an expression plasmid to evaluate the effect on IRF3 upon MERS-CoV infection. One can also utilize phospho tag gels for such experiments.
As the authors notice only very modest effect of MERS-CoV kinetics in bat cells that have reduced expression of IRF3 (Fig 5), this would suggest that some other factors in RIG-I dependent signaling cascades may also be involved as well? Has authors considered knocking down or studying the effect of RIG-I, MDA5, kinase activation TBK1 and IKK1 in this process?
Interestingly, in Fig 5 authors notice no difference MERS-CoV titers nor in IFNbexpression levels in MRC5 cell lines upon IRF3 knockdown, this likely provide evidence that virus is likely acting at some other level of the pathway. Furthermore, as MERS-CoV IFN antagonist protein 4a (Niemeyer et al, JVI 2013) is previously described to suppress IFN responses, is it possible that observed differences by authors is due to inability of IFN antagonists to suppress IFN responses in bat cells versus human cells?
The discussion need to be more refined to appropriately describe the findings and IFN responses in human versus bat cells should be elaborated.
Author Response
We would like to thanks the reviewer for their very helpful and thorough review. We appreciate the time this valuable service takes.
Reviewer's comments and our response (in italics):
In this work by Banerjee et. al. examine the antiviral activity of Middle East
Respiratory syndrome (MERS) coronavirus (CoV) infection in bat cells. Here,
the authors demonstrate that MERS-CoV grew to slightly higher titers in
human cells which is related to their capacity to suppress interferon (IFN)
expression albeit to a lesser extent in bat cells. Through this work, they
demonstrate that IRF3 plays a critical role in mediating IFN response during
MERS-CoV infection.
The manuscript examines an important aspect of MERS-CoV infection and is
well written but the major critique of this manuscript is that authors only
observe modest effect with MERS-CoV growth kinetics and IFN responses in
human vs bat cells and authors fail to decipher what is the major contributing
factor, for example why they see less suppression of IRF3 in bat cells in
comparison to human cells. Thus, a role for bat IRF3 in host response to
MERS-CoV infection or why it may be differs from human remains unclear. In
addition, strong data supporting the conclusion in the abstract that bat cells are
resistant to MERS-CoV mediated interferon responses and the biological
relevance of this function has not been provided.
Authors’ response: To identify why bat IRF3 is resistant to MERS-CoV-mediated shutdown, we attempted to perform a kinome analysis to look at the phosphorylation levels and phosphorylation sites in bat transcription factors, which would have enabled us to identify the kinases that are activated in bat cells, relative to human cells. Since IRF3 phosphorylation is associated with interferon production and subsequent interferon signaling, studying the kinases that activate IRF3 will provide us with a clearer picture of why bat IRF3-mediated signaling is resistant to shutdown by MERS-CoV. We may also be able to identify alternate/additional kinases that activate bat IRF3. However, we were limited to a human array to perform the kinome analysis and we did not obtain reliable data from bat samples (mock and infected). We will repeat this study when a kinome array specific for big brown bats, or a closely related bat species becomes available. In this study, we have highlighted the important role played by IRF3 in interferon production, both in response to double-stranded RNA and MERS-CoV. We also provide evidence for the first time that MERS-CoV cannot inhibit interferon production and signaling in bat cells, likely due to the resistance of IRF3-mediated signaling to MERS-CoV-mediated shutdown.
Some major concerns:
The authors only examine the effect of IRF3 but what about the effect on IFN
signaling pathway, such as effect on IFN signaling via the JAK/STAT pathway
in the two cell lines. What is effect on expression of IFN stimulated genes.
Authors’ response: As mentioned by the reviewer, MERS-CoV has been shown to suppress IFN stimulated genes (ISGs) that are activated via the interferon signaling pathway in human cells [5,6].
In our study, we have looked at the expression of OAS1, a key antiviral ISG. MERS-CoV infection induced the expression of OAS1 in bat cells and not in human cells (Fig 5C). This is consistent with the activation of IFN beta in MERS-CoV infected bat cells and not in human cells (Fig 5B). Thus, IFN signaling, and likely the JAK/STAT pathway are resistant to subversion by MERS-CoV proteins in bat cells, unlike human cells. The possibility of alternate pathways that can stimulate ISGs in bat cells remains but primary OAS1 expression in MERS-CoV infected bat cells is mediated by IRF3 (via interferon production and signaling) as observed in experiments where IRF3 has been knocked down (Fig 5C).
Authors used different cell lines but not cell line that are defective in IFN
production such as Vero cells, will the virus grow better in Vero cells as
compared to IFN competent cells?
Authors’ response: The objective of our study was to identify if human and bat cells differed in their interferon responses to MERS-CoV. Thus, we decided to include cells that are interferon competent in this study. We have previously studied the replication kinetics of MERS-CoV in Vero cells [7]. MERS-CoV replicates slightly better in Vero cells (107TCID50/ml) [7]that are interferon deficient relative to MRC5 cells (> 106TCID50/ml). This further supports other published studies and our study that MERS-CoV can effectively inhibit interferon production and signaling in human cells (MRC5) and replicate to comparable levels as observed in interferon-deficient cells (Vero cells).
In Fig 3, the effect on IRF3 localization is hard to interpret. As the authors have
the known coding sequence for bat IRF3, it would be cleaner experiment if
authors clone bat IRF3 to generate an expression plasmid to evaluate the
effect on IRF3 upon MERS-CoV infection. One can also utilize phospho tag
gels for such experiments.
Authors’ response: This is a good idea. However, we have struggled with plasmid transfection efficiencies in bat cells. We have attempted electroporation, but the process stimulates several ISGs. We are in the process of attempting to knock-in (using CRISPR/Cas9) a tag at the N-terminal end of endogenous bat IRF3. This however is challenging and will be part of a future study.
As the authors notice only very modest effect of MERS-CoV kinetics in bat
cells that have reduced expression of IRF3 (Fig 5), this would suggest that
some other factors in RIG-I dependent signaling cascades may also be
involved as well? Has authors considered knocking down or studying the
effect of RIG-I, MDA5, kinase activation TBK1 and IKK1 in this process?
Authors’ response: We have not attempted to knock down RIGI or MDA5 in this study. However, we did attempt to perform a kinome analysis on human and bat cells to identify phosphorylation sites on transcription factors and kinases that are activated. The results were not conclusive as we were limited to using a human kinome array. See first response to Reviewer 3 for details.
Interestingly, in Fig 5 authors notice no difference MERS-CoV titers nor in
IFNb expression levels in MRC5 cell lines upon IRF3 knockdown, this likely
provide evidence that virus is likely acting at some other level of the pathway.
Furthermore, as MERS-CoV IFN antagonist protein 4a (Niemeyer et al, JVI
2013) is previously described to suppress IFN responses, is it possible that
observed differences by authors is due to inability of IFN antagonists to
suppress IFN responses in bat cells versus human cells?
Authors’ response: The reviewer is correct. MERS-CoV likely inhibits IRF3 activation by disrupting the activation process via kinases such as TBK1. Alternatively, MERS-CoV could also de-phosphorylate IRF3 to inactivate IRF3. This remains to be tested in human and bat cells.
The reviewer is also correct in mentioning that the differences we observe is due to the inability of MERS-CoV proteins to suppress IFN responses in bat cells. This is the crux of our manuscript and it reiterates the message we are trying to put forward with this manuscript.
The discussion need to be more refined to appropriately describe the findings
and IFN responses in human versus bat cells should be elaborated
Authors’ response: We have included statements and references to other articles that have shown that MERS-CoV can inhibit an interferon response in human cells.
References
1. Banerjee, A.; Rapin, N.; Bollinger, T.; Misra, V. Lack of inflammatory gene expression in bats: a unique role for a transcription repressor. Sci Rep 2017, 7, 2232, doi:10.1038/s41598-017-01513-w.
2. Pavlovich, S.S.; Lovett, S.P.; Koroleva, G.; Guito, J.C.; Arnold, C.E.; Nagle, E.R.; Kulcsar, K.; Lee, A.; Thibaud-Nissen, F.; Hume, A.J., et al. The Egyptian Rousette Genome Reveals Unexpected Features of Bat Antiviral Immunity. Cell 2018, 173, 1098-1110 e1018, doi:10.1016/j.cell.2018.03.070.
3. Banerjee, A.; Falzarano, D.; Misra, V. Caution: choice of fixative can influence the visualization of the location of a transcription factor in mammalian cells. Biotechniques 2018, 65, 65-69, doi:10.2144/btn-2018-0060.
4. Banerjee, A.; Misra, V.; Schountz, T.; Baker, M.L. Tools to study pathogen-host interactions in bats. Virus Res 2018, 248, 5-12, doi:10.1016/j.virusres.2018.02.013.
5. Yang, Y.; Zhang, L.; Geng, H.; Deng, Y.; Huang, B.; Guo, Y.; Zhao, Z.; Tan, W. The structural and accessory proteins M, ORF 4a, ORF 4b, and ORF 5 of Middle East respiratory syndrome coronavirus (MERS-CoV) are potent interferon antagonists. Protein Cell 2013, 4, 951-961, doi:10.1007/s13238-013-3096-8.
6. de Wit, E.; van Doremalen, N.; Falzarano, D.; Munster, V.J. SARS and MERS: recent insights into emerging coronaviruses. Nat Rev Microbiol 2016, 14, 523-534, doi:10.1038/nrmicro.2016.81.
7. Banerjee, A.; Rapin, N.; Miller, M.; Griebel, P.; Zhou, Y.; Munster, V.; Misra, V. Generation and Characterization of Eptesicus fuscus (Big brown bat) kidney cell lines immortalized using the Myotis polyomavirus large T-antigen. J Virol Methods 2016, 237, 166-173, doi:10.1016/j.jviromet.2016.09.008.
Round 2
Reviewer 2 Report
Although author's did not perform experiments suggested, they have provided several references and reasoning in the discussion. Despite lack of few controls, these additional changes have adequately addressed this reviewer's questions.
The authors should clearly mention that IFN-a is not constitutively expressed in big brown bat cells. They mention IFN-a in line 521-524 (instead of 537-540 as stated in response to reviewer), but this explanation is insufficient.
Author Response
Although author's did not perform experiments suggested, they have provided several references and reasoning in the discussion. Despite lack of few controls, these additional changes have adequately addressed this reviewer's questions.
The authors should clearly mention that IFN-a is not constitutively expressed in big brown bat cells. They mention IFN-a in line 521-524 (instead of 537-540 as stated in response to reviewer), but this explanation is insufficient.
Authors’ response: we have clarified that we do not detect IFN alpha in big brown bat kidney cells (lines 544-550). We have also added a statement on studying the interaction of Pin1 and IRF3 in immune relevant cells.
Reviewer 3 Report
In the revised version of the manuscript authors have addressed some of the concerns but one of the major concerns still remains with justifying IRF3 as major player in observed differences of MERS-CoV infections in humans versus bat cells. My major critique is with data IRF3 presented in figures 3-5.
In Figure 3, the localization of IRF3 is hard to interpret as Red signal presumably representing IRF3 is also seen in nucleus in mock treated cells as well. The quantification of the nuclear/cytoplasmic ratios is also not convincing as authors only used n=5 and p values especially for MRC is border line significant. The authors need to provide more convincing immunofluorescence data. This data will be more meaningful if they can perform similar experiment in presence of MERS-CoV infection. The data presented in panel C is also not clear as the “higher band” that authors referring as an IRF3 phosphorylation band, authors should provide better image or include the whole blot rather than cutting off at the edge to clearly show presence of two separate bands. Also, if IRF3 is more nuclear then why overall there is less IRF3 in their nuclear extracts vs cytoplasmic extracts. Overall, the IRF3 Western blots provided throughout the manuscript are not clear. In Figure 4B, even with non-specific siRNA there is decrease in IRF3 expression and then in panel 4D, its not clear which band is IRF3 as Efk3 WT shows a very weak band and as the image is tilted its hard to judge whether that band is IRF3 or not and there are still weak bands present in CRISPR cell lines. The authors need to provide the whole image of blot and also may be qRT-PCR of IRF3 mRNA levels. As the knockout of IRF3 is questionable in these cell lines, maybe that’s why authors only see very modest difference between wt and CRISPR efk3 cell lines (p=0.046) in Figure 5C.
As IRF3 is one of the major highlights of the manuscript distinguishing MERS-CoV effect on innate immune responses in human versus bat cells, authors need to provide additional experiments or at least clean data for Figures 3-5.
Also, its not clear whether statistical significance was determined by using 2 independent samples or experiments or based on “n” in each experiment. As it if its only 2 samples that would not be provide appropriate statistical scoring unless its 2 independent experiments with each having 3-4 independent replicates.
Additionally, it will be of interest to include brief discussion explaining why MERS-CoV are less efficient in blocking IFN responses in bat cells, what it means with respect to virus evolution or is it function of IFN antagonist viral proteins in the two cell lines.
Author Response
In the revised version of the manuscript authors have addressed some of the concerns but one of the major concerns still remains with justifying IRF3 as major player in observed differences of MERS-CoV infections in humans versus bat cells. My major critique is with data IRF3 presented in figures 3-5. In Figure 3, the localization of IRF3 is hard to interpret as Red signal presumably representing IRF3 is also seen in nucleus in mock treated cells as well. The quantification of the nuclear/cytoplasmic ratios is also not convincing as authors only used n=5 and p values especially for MRC is border line significant. The authors need to provide more convincing immunofluorescence data. This data will be more meaningful if they can perform similar experiment in presence of MERS-CoV infection.
Authors’ response: We would indeed like to carry out these experiments with MERS-CoV infection, however, at the moment the mandated step of fixing the cells using 10% formalin prior to taking the samples out of containment level 3 does not allow us to carry out experiments with MERS-CoV. We have previously published that fixation using 10% formalin and other fixatives disrupt the cellular localization of IRF3 (Banerjee et al.; Biotechniques, 2018).
The data presented in panel C is also not clear as the “higher band” that authors referring as an IRF3 phosphorylation band, authors should provide better image or include the whole blot rather than cutting off at the edge to clearly show presence of two separate bands.
Authors’ response: We have now included whole blots for all our westerns blots in Appendix A.
Also, if IRF3 is more nuclear then why overall there is less IRF3 in their nuclear extracts vs cytoplasmic extracts. Overall, the IRF3 Western blots provided throughout the manuscript are not clear.
Authors’ response: We have now included whole blots for all our westerns blots in Appendix A. IRF3 is known to shuttle between the nucleus and the cytoplasm. Depending on the time after activation, IRF3 can be detected either in the nucleus or cytoplasm. Since phosphorylated IRF3 migrates to the cytoplasm, using a phosho-IRF3 antibody would be a better representation of the quantity of ‘activated’ IRF3 in the cytoplasm vs. nucleus. However, commercially available human phospho-IRF3 antibodies did not cross-react with IRF3 in bat cells that were treated with poly(I:C). This phenomenon is further complicated by the likely degradation of activated IRF3 by Pin1. However, our western blots clearly demonstrate the presence of post-translational modifications (high molecular weight bands) in poly(I:C) treated samples alone.
In Figure 4B, even with non-specific siRNA there is decrease in IRF3 expression and then in panel 4D, its not clear which band is IRF3 as Efk3 WT shows a very weak band and as the image is tilted its hard to judge whether that band is IRF3 or not and there are still weak bands present in CRISPR cell lines. The authors need to provide the whole image of blot and also may be qRT-PCR of IRF3 mRNA levels.
Authors’ response: The degradation of human IRF3 with non-specific siRNA is likely due to the action of Pin1. Pin1 selectively marks activated (phosphorylated) IRF3 for degradation. During our studies, we did not observe this in bat cells (discussed in the Discussion section). This observation has now developed into an independent project where we are looking at the negative regulation of IRF3 in human and bat cells.
The weak bands in the CRISPR KO cells are non-specific bands. We see them in all the blots for IRF3. The commercially available IRF3 antibody does detect non-specific bands. This has been reported by others and this can also be observed in the western blot displayed on the manufacturer’s website (Figure 1):https://www.abcam.com/irf3-antibody-epr2418y-ab68481.html#description_images_8
In addition to performing western blots, we have also confirmed the knockout of IRF3 in bat cells by sequencing the genomic locus for IRF3 and by performing functional studies using poly(I:C) (Figure 4D).
As the knockout of IRF3 is questionable in these cell lines, maybe that’s why authors only see very modest difference between wt and CRISPR efk3 cell lines (p=0.046) in Figure 5C.
Authors’ response: As mentioned above, we have confirmed the knockout of IRF3 using functional studies with poly(I:C) and by sequencing the genomic locus for IRF3 in bat cells. There are likely unidentified factors that control virus replication in addition to IRF3-mediated signaling. The KO cells will enable us to identify these factors in the future.
As IRF3 is one of the major highlights of the manuscript distinguishing MERS-CoV effect on innate immune responses in human versus bat cells, authors need to provide additional experiments or at least clean data for Figures 3-5.
Authors’ response: We have now included whole blots for all our westerns blots in Appendix A.
Also, its not clear whether statistical significance was determined by using 2 independent samples or experiments or based on “n” in each experiment. As it if its only 2 samples that would not be provide appropriate statistical scoring unless its 2 independent experiments with each having 3-4 independent replicates.
Authors’ response: We have used ‘biological replicates’ to distinguish our replicates from ‘technical replicates’. All experiments were carried out in ‘n’ number of independent experiments and each experiment had 3 replicates.
Additionally, it will be of interest to include brief discussion explaining why MERS-CoV are less efficient in blocking IFN responses in bat cells, what it means with respect to virus evolution or is it function of IFN antagonist viral proteins in the two cell lines.
Authors’ response: We have added a few statements on possible future studies to address these questions (lines 585-594).
Round 3
Reviewer 3 Report
The clarification provided by authors is satisfactory. To improve this manuscript for publication, authors are recommended to provide better immunofluorescence images in Figure 3 and quantification with at least 50 cells that provide substantial data to support nuclear localization of IRF3 upon polyI:C treatment. As shown by authors human IRF3 antibody is cross-reactive against bat IRF3, so that should be sufficient to obtain better quality data for publication. One would expect the ‘activated IRF3’ to be present predominantly in the nucleus not in the cytoplasm, this will further suggest that IRF3 in bat cells is responsive to dsRNA and presumably the virus infection and this also support authors argument that IRF3 in bat cells is important for restricting MERS-CoV infection.
Author Response
Reviewer 3
The clarification provided by authors is satisfactory. To improve this manuscript for publication, authors are recommended to provide better immunofluorescence images in Figure 3 and quantification with at least 50 cells that provide substantial data to support nuclear localization of IRF3 upon polyI:C treatment. As shown by authors human IRF3 antibody is cross-reactive against bat IRF3, so that should be sufficient to obtain better quality data for publication. One would expect the ‘activated IRF3’ to be present predominantly in the nucleus not in the cytoplasm, this will further suggest that IRF3 in bat cells is responsive to dsRNA and presumably the virus infection and this also support authors argument that IRF3 in bat cells is important for restricting MERS-CoV infection.
Authors’ response:We have included a higher quality image (400 dpi). The immunofluorescence data is representative of five different microscopic fields and not ‘five cells’. The data includes several cells that were observed and measured in n=5 fields, as mentioned in the figure legend.
Although we and others do observe IRF3 ‘predominantly’ in the nucleus after poly(I:C) treatment, this can vary based on the timing of the experiment and the amount of stimulus (polyIC). ‘Activated IRF3’ or phosphorylated IRF3 is a target for Pin1-mediated degradation and Pin1 causes IRF3 to localize in the cytoplasm [1]. However, in our immunofluorescence images, IRF3 can be observed in the nucleus of human and bat cells. The localization kinetics may differ in individual cells based on the timing of poly(I:C) delivery in these cells. Not all cells in a tissue culture plate would take up poly(I:C) at the same rate, which would in turn affect IRF3-Pin1 interactions and subsequent detection of IRF3 in the nucleus/cytoplasm of these cells.
In our opinion, knockdown of and CRISPR knockout of IRF3 in bat cells provides strong and convincing evidence that IRF3-mediated signaling plays a significant role in limiting MERS-CoV propagation in bat cells. For future follow-up studies, we are considering an RNAseq experiment to identify alternate pathways that may be involved in antiviral signaling in bat cells. We are also exploring the possibility of IRF3-independent signaling in bat cells as has been demonstrated for other mammalian cells [2].
Other authors’ notes:We have updated the title of the manuscript. We have replaced ‘response’ with ‘regulatory’. Interferon regulatory factor 3 (IRF3) is a more common and accepted name.
References
1. Saitoh, T.; Tun-Kyi, A.; Ryo, A.; Yamamoto, M.; Finn, G.; Fujita, T.; Akira, S.; Yamamoto, N.; Lu, K.P.; Yamaoka, S. Negative regulation of interferon-regulatory factor 3-dependent innate antiviral response by the prolyl isomerase Pin1. Nat Immunol 2006, 7, 598-605, doi:10.1038/ni1347.
2. DeWitte-Orr, S.J.; Mehta, D.R.; Collins, S.E.; Suthar, M.S.; Gale, M., Jr.; Mossman, K.L. Long double-stranded RNA induces an antiviral response independent of IFN regulatory factor 3, IFN-beta promoter stimulator 1, and IFN. J Immunol 2009, 183, 6545-6553, doi:10.4049/jimmunol.0900867.